# Evaluation of MITgcm-based ocean reanalyses for the Southern Ocean

Yoshihiro Nakayama[1], Alena Malyarenko[2,3], Hong Zhang[4], Ou Wang[4], Matthis Auger[5,6], Yafei Nie[7], Ian Fenty[4], Matthew Mazloff[8], Köhl Armin[9], and Dimitris Menemenlis[4]

[1]Institute of Low Temperature Science, Hokkaido University, Sapporo, Hokkaido, Japan.
[2]Te Kura Aronukurangi | School of Earth and Environment, Te Whare Wānanga o Waitaha | University of Canterbury, Ōtautahi | Christchurch, Aotearoa | New Zealand
[3]Te Puna Pātiotio | Antarctic Research Centre, Te Herenga-Waka Victoria University of Wellington, Aotearoa | New Zealand
[4]Jet Propulsion Laboratory, California Institute of Technology, Pasadena, CA, USA.
[5]Institute for Marine and Antarctic Studies, University of Tasmania, Hobart, Australia
[6]The Australian Centre for Excellence in Antarctic Science, University of Tasmania, Hobart, Australia
[7]School of Atmospheric Sciences, Sun Yat-Sen University, and Southern Marine Science and Engineering Guangdong Laboratory (Zhuhai), Zhuhai, China
[8]Scripps Institution of Oceanography, University of California San Diego, La Jolla, CA, USA
[9]Center for Earth System Research and Sustainability (CEN), Universität Hamburg, Hamburg, Germany

**Correspondence:** Y. Nakayama (Yoshihiro.Nakayama@lowtem.hokudai.ac.jp)

**Abstract.** Global and basin-scale ocean reanalyses are becoming easily accessible and are utilized widely to study the Southern Ocean. Yet, such ocean reanalyses are optimized to achieve the best model-data agreement for their entire model domains and their ability to simulate the Southern Ocean requires investigations. Here, we compare several Massachusetts Institute of Technology general circulation model (MITgcm)-based ocean reanalyses (ECCOv4r5, ECCO LLC270, B-SOSE, and GECCO3) for the Southern Ocean. For the open ocean, the simulated time-mean hydrography and ocean circulation are similar to observations. The MITgcm-based ocean reanalyses show Antarctic Circumpolar Currents (ACC) measuring approximately 149±11Sv. The simulated 2 °C isotherms are located in positions similar to the ACC and roughly represent the southern extent of the current. Simulated Weddell and Ross Gyre strengths are 51±11Sv and 25±8Sv, respectively, consistent with observation-based estimates. However, our evaluation finds that the time evolution of the Southern Ocean is not well simulated in these ocean reanalyses. While observations showed little change in open-ocean properties in the Weddell and Ross Gyres, all simulations showed larger trends, most of which are excessive warming. For the continental shelf region, all reanalyses are unable to reproduce observed hydrographic features, suggesting that simulated physics determining on-shelf hydrography and circulation is not well represented. Nevertheless, ocean reanalyses are valuable resources and can be used for generating ocean lateral boundary conditions for regional high-resolution simulations. We recommend that future users of these ocean reanalyses pay extra attention if their studies target open-ocean Southern Ocean temporal changes or on-shelf processes.

# 1 Introduction

The Southern Ocean is a critical component of the Earth's climate system (e.g., Gille, 2002; Hellmer et al., 2012; Rintoul, 2018). The Antarctic Circumpolar Current (ACC), Ross Gyre (RG), and Weddell Gyre (WG) circulations are involved in the advection of warm ocean waters towards the Antarctic ice shelves (e.g., Orsi et al., 1999; Ryan et al., 2016; Nakayama et al., 2018). The formation of sea ice close to the Antarctic continent leads to the production of high salinity shelf water that sinks to the ocean's bottom, forming the Antarctic Bottom Water (AABW) — the coldest and densest water mass in the world's oceans (e.g., Jacobs et al., 1970; Foldvik et al., 2004; Williams et al., 2008; Ohshima et al., 2013). The Southern Ocean also serves as a significant sink for anthropogenic carbon, playing a crucial role in carbon cycling (e.g., Takahashi et al., 2012; Sallée et al., 2023; Williams et al., 2023). However, our knowledge of the Southern Ocean is still limited due to a lack of observations in the challenging polar conditions.

One approach to address the lack of observations is to create global, circum-Antarctic, or regional ocean models. In recent decades, several researchers have developed coupled sea ice, ice shelf, and ocean models, allowing us to study the interplay between large-scale ocean circulation, ice shelves, and sea ice on a global or circum-Antarctic scale (e.g., Timmermann et al., 2009; Nakayama et al., 2013; Kusahara and Hasumi, 2014; Dinniman et al., 2015; Schodlok et al., 2015; Mathiot et al., 2017; Kiss et al., 2020). To study processes over the Antarctic continental shelves, other researchers develop regional simulations (e.g., Dinniman et al., 2011; Schodlok et al., 2012; Gwyther et al., 2014; Nakayama et al., 2014a; Jourdain et al., 2017; Nakayama et al., 2017; Mack et al., 2019; Naughten et al., 2019; Nakayama et al., 2019, 2021a; Naughten et al., 2022). Regional models can yield a superior agreement between model and observations because (1) regional simulations can achieve high spatial and vertical resolutions and (2) model parameters can be tuned for the study regions. Thus, regional simulations can help better interpret observed on-shelf processes. For example, several studies have investigated on-shelf intrusions of warm Circumpolar Deep Water (CDW) or the formation of AABW (Assmann et al., 2013; Nakayama et al., 2018; Mensah et al., 2021). Such models can also be used to identify ideal locations for ocean observations (e.g., deployment locations of Argo floats). Nonetheless, constructing, evaluating, and improving these regional ocean models requires a significant amount of time and effort.

Another approach is to utilize existing ocean reanalyses, which are convenient and becoming an increasingly trusted tool. Over the past decades, following extensive developmental efforts, several ocean reanalyses have become available. Previously Uotila et al. (2019) have analyzed the mean states of ten ocean reanalysis products, including GECCO2 (a prior version of the GECCO3 reanalysis discussed below) for the Arctic and Antarctic focusing on sea ice and upper-ocean hydrography. Here, we will evaluate some recent ocean reanalyses based on the Massachusetts Institute of Technology general circulation model (MITgcm) and made available by the Estimating the Circulation and Climate of the Ocean consortium, including ECCO version 4 release 5 (ECCOv4r5; Forget et al., 2015), ECCO Latitude-Longtitude-polar Cap 270 (ECCO-LLC270; Zhang et al., 2018), the German contribution of the ECCO project version 3 (GECCO3; Köhl, 2020), and the Biogeochemical-Sea ice-Ocean State Estimate iterations 105 and 139 (B-SOSE; Mazloff et al., 2010; Verdy et al., 2017). As these are ocean reanalyses, researchers in various fields, especially non-ocean modelers assume that these model outputs replicate past changes and variability in the

Southern Ocean to some extent. It is convenient for researchers because they can download one of these ocean reanalyses and use the same model output repeatedly for their different research. For example, they can use the same code to read the model data, conduct model-data comparisons, or test their hypothesis to strengthen their scientific claims.

Ocean reanalyses are generated by minimizing a cost function (weighted sum of model-data difference squared) by optimizing model parameters, including atmospheric forcing, ocean mixing parameters, and initial conditions (Forget et al., 2015; Zhang et al., 2018). All these MITgcm-ECCO assimilations (ECCOv4r5, ECCO-LLC270, B-SOSE, and GECCO) are unique in that they use the model as a hard constraint over many years. Long assimilation windows are chosen to yield the advantage of having closed budgets. This means that ECCO products assume the model is perfect and chaos does not impact the large scale over long time scales. In contrast, other groups nudge the model on daily to weekly time scales, keeping their models better on track with the observations. Challenges arise when applying this framework to simulate the Southern Ocean. First, the observations used to constrain ocean models are primarily concentrated in lower-latitude, ice-free regions, and in summertime (Newman et al., 2019). Even if the cost is reduced for the global ocean, it is possible that the ocean state in the Southern Ocean remains suboptimal. Second, except for ECCOv4r5, most ocean reanalyses do not explicitly model ice-shelf cavities (Forget et al., 2015; Zhang et al., 2018; Mazloff et al., 2010; Verdy et al., 2017; Köhl, 2020). We also note that ocean state estimates development takes a long time and model bathymetry can be outdated for the Antarctic continental shelf regions. Third, our optimization framework uses long assimilation windows. This means that we do not have controllability of the mesoscale dynamics, which may modulate the hydrography and circulation of the Southern Ocean (Ito and Marshall, 2008; Ruan et al., 2017). Fourth, polar scientists care about tiny changes in temperature or thermocline depth. Such changes may not be a priority for ocean reanalysis developers. Finally, while developers evaluate ocean reanalyses, there has been no cross-comparison of these MITgcm-based ocean reanalysis products and observations focusing on the entire Southern Ocean such as plotting sections and comparing time series against observations despite some regional attempts for reanalyses intercomparisons (e.g., Chen et al., 2023; Bailey et al., 2023; Cerovečki and Haumann, 2023).

In this study, we evaluate Southern Ocean circulation in the four, aforementioned ECCO reanalyses. We compare model output from these reanalyses to existing observations in the open ocean and over the continental shelves, examining how well they simulate the mean state, variability, and trends. We aim to provide guidelines for researchers on what they can and cannot do with these ocean reanalyses. We also would like to (1) make recommendations on how much these ocean reanalyses can be trusted for the purposes of creating initial conditions and lateral forcing for regional simulations and (2) suggest how we can improve the model-data agreement in the Southern Ocean for the existing ocean reanalyses.

## 2   Ocean reanalyses and datasets

The four ECCO reanalyses that will be evaluated in this study are listed in Table 1 and briefly described below.

## 2.1 ECCOv4r5

ECCO Version 4 release 5 (v4r5) is ECCO's latest global ocean state estimate covering the period of 1992–2019 (Table 1). This product is an updated version of the solution described in Forget et al. (2015). ECCOv4r5 uses the LLC90 grid, which has a nominal horizontal grid spacing of 1° between 70°S and 57°N. South of 70°S, the LLC90 configuration uses a bipolar grid. Horizontal grid spacing is approximately 40 km along the Antarctic Coast. Details of the previous releases can be found in Forget et al. (2015). ECCOv4r5 uses hourly MERRA2 for the initial guess of atmospheric forcing.

We also highlight two important improvements related to polar regions compared to the previous versions (ECCOv4r4 and ECCO-LLC270). First, release 5 includes static ice shelf cavities. Ice shelf melt rates are calculated using a three-equation model (Hellmer and Olbers, 1989; Jenkins, 1991; Holland and Jenkins, 1999), with exchange coefficients of heat and salt at the ocean-ice shelf boundary being adjusted through optimization. Model bathymetry around Antarctica is from The International Bathymetric Chart of the Southern Ocean (IBCSO) Version 1.0. (Arndt et al., 2013). Ice shelf thickness is based on Bedmap-2 (Fretwell et al., 2013). Together, IBCSO bathymetry and Bedmap-2 ice-shelf thickness are used to update geometry around Antarctica. Second, starting from version 4 release 5, sea-ice thermodynamics are included in adjoint-model sensitivity computations achieving higher model data agreement for sea ice extent and variability as discussed below.

## 2.2 ECCO-LLC270

ECCO-LLC270 is a higher-resolution, global ECCO ocean state estimate covering 1992–2017 (Table 1). This reanalysis is an updated version of the solution described in Zhang et al. (2018). ECCO-LLC270 uses the LLC270 grid, which has a nominal horizontal grid spacing of 1/3° between 70°S and 57°N. South of 70°S, the LLC270 configuration also uses a bipolar grid with a horizontal grid spacing of approximately 15 km along the Antarctic Coast. ECCO-LLC270 does not include ice shelf cavities. Model bathymetry is based on the Global Sea Floor Topography from Satellite Altimetry and Ship Depth Soundings (Smith and Sandwell, 1997), which is problematic in some regions around Antarctica, e.g., in the Amundsen Sea. In this study, we evaluate Iteration 50 of ECCO-LLC270, which is similar to iteration 42 used in ECCO-Darwin (Carroll et al., 2020). ECCO-LLC270 uses 6 hourly ERA-interim for the initial guess of atmospheric forcing.

## 2.3 B-SOSE

B-SOSE is a circum-Antarctic regional ocean state estimate developed by the Scripps Institution of Oceanography (Table 1). The iteration 139 we used covers the period 2013-2021. The analysis model domain extends from 30°S to 78°S. The zonal grid spacing is 1/6°. B-SOSE does not include ice shelf cavities. Model bathymetry is based on ETOPO1 (Amante and Eakins, 2009). Details can be found in Mazloff et al. (2010) and Verdy et al. (2017). B-SOSE includes biogeochemistry, but we do not evaluate this component in this manuscript. We also analyze the iteration 105, which covers 2008-2012. B-SOSE iteration 105 has a zonal grid spacing of 1/3° and has been used widely for Southern Ocean studies. B-SOSE iteration 139 uses hourly ERA5 for the initial guess of atmospheric forcing (Verdy et al., 2017).

## 2.4 GECCO3

GECCO3 (Köhl, 2020) is a global ocean state estimate developed by Universität Hamburg, Germany (Table 1). This simulation covers the period 1948-2018. GECCO3 uses the bathymetry and grid of the MPI-ESM with a quasi-uniform resolution of 0.4°. GECCO3 does not include ice shelf cavities. Model bathymetry is from the MPI-HR configuration of the MPI-ESM climate model which is based on ETOPO5 (National Geophysical Data Center 1988) (Jungclaus et al., 2013). In this comparison, we use the results of GECCO3S6m with surface salinity relaxation with a relaxation time scale of 6 months. The detail can be found in Köhl (2020). GECCO3 uses 6 hourly NCEP/NCAR Reanalysis 1 for the initial guess of atmospheric forcing.

## 2.5 Sea ice observations used for reanalysis evaluation

We use the monthly sea ice concentration observational data obtained from the National Oceanic and Atmospheric Administration/National Snow and Ice Data Center (NOAA/NSIDC) Climate Data Record of Passive Microwave Sea Ice Concentration, version 4 (CDR) (Meier et al., 2021). We use two observational sea ice thickness references. The first one, released by the European Space Agency Sea Ice Climate Change Initiative (SICCI), provides sea ice thickness records from June 2002 to April 2017 and has a spatial resolution of 50 km. The uncertainty of this dataset is considerably large (Wang et al., 2022; Hou et al., 2024). The second sea ice thickness data is provided by the Laboratoire d'Etudes en Géophysique et Océanographie Spatiales (LEGOS) (Garnier et al., 2021, 2022) with a 12.5 km resolution. The LEGOS sea ice thicknesses benefit from improved snow depth measurements on sea ice, which are obtained by altimeters on board the satellites SARAL (Ka-band) and CS2 (Ku-band). To avoid the effect of snowpack property variabilities (e.g., density and grain size) on retrieval accuracy, the LEGOS sea ice thickness data is only available in winter months (from May to October) and covers from May 2013 to October 2018.

## 2.6 Ocean observations used for reanalysis evaluation

We use multiple datasets for the ocean reanalysis evaluation. We use the World Ocean Circulation Experiment (WOCE) Hydrographic Sections S4P and SR04 for the Ross and Weddell Sea, respectively. The S4P section was observed in 1992 (WOCE), 2011, and 2018 (Talley, 2007; Purkey and Johnson, 2013; Purkey et al., 2019). The SR04 Section was observed in 1989, 1996 (WOCE Section 23), 2005, and 2010 (Fahrbach et al., 1991, 2004; Talley, 2007; Rücker van Caspel, 2016). We also use World Ocean Atlas 2018 as a reference dataset for the entire Southern Ocean (Locarnini et al., 2018; Zweng et al., 2019). A Mean Dynamic Topography (MDT) obtained from satellite altimetry observations that include ice-covered regions (Armitage et al., 2018) is displayed to qualitatively compare with reanalysis stream functions. This MDT is referenced to the GOCO05c combined gravity field model (Fecher et al., 2017).

## 3  Results

### 3.1  Ocean bathymetry and ice shelves

The ocean reanalyses use ocean bathymetry derived from three different products (Fig. 1). We find obvious differences over the continental shelves. ECCO-LLC270 and GECCO3 use the older bathymetry product, which results in the absence of deep submarine glacial troughs (approximately 500 meters) connecting the open water and ice shelf cavities in the Amundsen Sea, Bellingshausen Sea, and western Ross Sea (Fig. 1). B-SOSE uses newer ETOPO1 bathymetry (Amante and Eakins, 2009), which is substantially improved compared to ECCO-LLC270 and GECCO3. However, there are still small differences between ECCOv4r5 and B-SOSE, including the absence of detailed structures of submarine glacial troughs in the Amundsen and Bellingshausen Seas. B-SOSE also cuts out the southernmost part of the Ross Sea continental shelf due to its horizontal gridding configuration. ECCOv4r5, the newest ocean reanalyses, utilizes the best available bathymetric data and has ice shelf cavities. For the open ocean region, the ocean bathymetries of these analyses are similar.

### 3.2  Sea ice

We first evaluate the sea ice area (SIA), which is the sum of grid cell areas multiplied by their respective sea ice concentrations, provided the concentration is at least 15%. For ECCOv4r5, ECCO-LLC270, B-SOSE (iter105), and GECCO3, the monthly SIA time series are similar to observations (Fig. 2a). All the reanalyses well capture the seasonal cycles of SIA, with minimum and maximum in February and September, respectively (Fig. 2d). To further focus on SIA in winter and summer, we plot detrended monthly anomalies in September and February (Figs. 2b-c). All reanalyses show near-realistic anomaly evolutions in September, but only ECCOv4r5, ECCO-LLC270, and GECCO3 performed well in February, while B-SOSE has significantly larger variation than the observational data. The standard deviation of February and September anomalies of each reanalysis are quantitatively compared to the observations (Table 2). The ECCOv4r5 and ECCO LLC270 show the most realistic interannual variations in February and September respectively, with the former also having the highest correlation with the observation in both months. Overall, the ECCOv4r5 performs the best in the comparison, which likely indicates the effectiveness of sea-ice thermodynamics adjoint-model sensitivity computations. Comparing reanalyses with long integration (ECCOv4r5, ECCO-LLC270, and GECCO), ECCOv4r5 and ECCO-LLC270 show good agreement in February at a similar level, suggesting that sea-ice thermodynamics adjoint sensitivity computation is not improving summer sea ice extent.

We also conduct the same analysis for sea ice volume (SIV) (Fig. 3). For the seasonality, only ECCO LLC270 shows minimum and maximum in February and September respectively, inconsistent with the observations (Fig. 3d). The maximum SIV month of other reanalyses all delayed to October, the same issue was found in the Global Ice-Ocean Modeling and Assimilation System (Liao et al., 2022), the ECMWF Ocean ReAnalysis System 5 (Nie et al., 2022) and many Coupled Model Intercomparison Project Phase 6 climate model results (Hou et al., 2024). All reanalyses underestimate the SIV throughout the year, but the B-SOSE (iter139) is the closest to the observations. For the interannual variability of sea ice volumes in September and February, all state estimates show similar temporal variability, and their variabilities are similar to that of sea ice extent (Figs. 2b-c and 3b-c).

### 3.3 Large-scale ocean state

#### 3.3.1 Large-scale hydrographic structure

We select 550 m to compare simulated outputs and the World Ocean Atlas (WOA) dataset, as this is a typical depth for CDW to travel in the open ocean before crossing the continental shelf break towards the Antarctic ice shelves (Nakayama et al., 2018).
In the WOA dataset (Locarnini et al., 2018; Zweng et al., 2019), the annual mean 1°C isotherm and 34.6 isohaline encompass the Southern Ocean, separating warm and cold water masses (Figs. 4e,j). Similarly, the 1°C isotherms and 34.6 isohalines of all ocean reanalyses are located at similar positions, indicating that all four reanalyses generally simulate similar hydrographic structures (Figs. 4 and S1). Inside the 1°C isotherms, WOA shows two local temperature minima in the Weddell Sea and Ross Sea (Figs. 4e,j), which are also present in all four ocean reanalyses. Inside the 34.6 isohaline, observations show a local maximum in the Ross Sea (Figs. 4e,j), and such structures are also simulated in ECCOv4r5, ECCO-LLC270, and B-SOSE (iterations 105 and 139).

We compare the February WOA mean to the February mean state of the reanalyses (Figs. 5 and S2) because the WOA dataset is biased towards summer observations. All reanalyses represent the surface temperature and salinity values well. The position of isolines and isohalines agrees between the reanalyses and matches the WOA for the Southern Ocean (Fig. 5). Close to the continent, the Weddell Sea matches the temperature and salinity values particularly well except for B-SOSE: the cold and relatively salty surface water mass is a feature of ECCOv4r5, ECCO-LLC270, and GECCO3. However, the sector between the Antarctic Peninsula and the Ross Sea has a fresh surface bias within the coastal current, confined to the top layer. Among these model outputs, B-SOSE (iter 139) shows the strongest freshening (Fig. 5h). We hypothesize that the fresh bias is related, in part, to the choice of freshwater flux from the Antarctic continent. In ECCOv4r5, ice sheet melt is computed, and iceberg calving is prescribed (Hammond and Jones, 2017). In ECCO-LLC270, both melt and calving are prescribed (see Feng et al., 2021, Fig. S1b). In B-SOSE, melt and calving are prescribed, also based on Hammond and Jones (2017), and the fresh bias is partially exacerbated by too strong a sea ice seasonal cycle (i.e., too much formation in winter). GECCO3 does not prescribe any freshwater discharge around Antarctica. Local sea ice melting likely also contributes to the fresh cap creation during spring, which is not vertically mixed in summer. It is likely that the warm SST bias in this region is also caused by the fresh bias: stratified water column does not promote vertical mixing of the summer warming.

We also compare mixed layer depths (MLD) because ocean reanalyses are widely used for physical oceanography and biogeochemistry studies (Fig. 6). MLD is defined as the depth level where the density difference from the surface exceeds 0.03 kg m$^{-1}$ following Sallée et al. (2010). We select September and February, the same time as sea ice concentration and volume. These months coincide with or are close to the MLD maximum and minimum. In September, observed MLD is located deep (about 500 m) at lower latitude regions ($\sim$50°S) in the Pacific sector. Similar patterns are simulated for all ocean reanalyses. We also find several spots with deep MLD along the Antarctic coasts, which are likely representing formation regions of AABW in the WOA. However, none of the ocean reanalyses show such deep MLD along the Antarctic coast. In February, MLDs become shallower for all locations but maintain deep MLD (about 100 m) regions at lower latitudes ($\sim$50°S). Such patterns are mostly present in all reanalyses but simulated MLDs are biased shallow in all ocean reanalyses presented in this study. For example,

February spatial mean MLDs for the region south of 60°S are 26 m, 24 m, 15 m, 17 m, and 49 m for ECCOv4r5, ECCO LLC270, B-SOSE (iter139), GECCO3, and WOA, respectively.

We show time series of MLD at the center of the Weddell and the Ross Gyres as ocean reanalyses tend to show better agreement for the open water regions (Fig. 7). All ocean reanalyses show similar seasonal cycles with maximum and minimum in December/January and September/October, respectively (Figs, 7c-d). Simulated MLDs show small variability, and the

210 standard deviations of simulated MLDs are about or less than 30 m. Standard deviations become large between October and December during the shallowing of MLDs. The time series of the simulated MLD at the center of the Weddell Gyre shows larger interannual variability, while the time series of simulated MLD at the center of the Ross Gyre shows smaller interannual variability (Figs. 7a-b).

We also calculate the Root Mean Square Error (RMSE) of potential temperature and salinity at depth levels of 500 m, 2000

215 m, and 4000 m (Fig. 8). RMSE is calculated by interpolating each ocean state estimate output onto a common horizontal grid with a 1° resolution and computing the difference to WOA potential temperature and salinity climatology fields. At 500 m, most ocean reanalyses show stable RMSE indicating that ocean reanalyses have nearly constant error throughout the model simulations. ECCOv4r5 and ECCO-LLC270 perform better, maintaining similar RMSE during model integration over approximately 30 years. In contrast, GECCO shows a rapid increase in RMSE within the first 10 years, after which

RMSE stabilizes or slightly decreases (Figs. 8a,d). B-SOSE exhibits the steepest increase in RMSE among the ocean analyses presented here. At the deeper levels (2000 m and 4000 m), all ocean reanalyses show increases in RMSE (except GECCO3 salinity) at similar rates (Figs. 8b,c,e,f). This consistent increase in RMSE may reflect the fact that all ocean reanalyses struggle to accurately simulate Antarctic Bottom Water (AABW) production, thereby misrepresenting key ocean processes leading to similar responses in all ocean reanalyses' RMSE (discussed further below).

### 3.3.2 Large-scale Ocean Circulation

Recent observations estimated the ACC strength to be 140-170 Sv using advanced technologies (Donohue et al., 2016; De Verdière and Ollitrault, 2016; Xu et al., 2020). Other canonical estimates are 130-140 Sv (Whitworth III et al., 1982; Whitworth, 1983; Whitworth and Peterson, 1985; Koenig et al., 2014). For ocean reanalysis, we also calculate the time mean ACC strengths by determining the flux through the Drake Passage for each simulated period. The mean ACC strengths for the

230 ECCOv4r5, ECCO-LLC270, B-SOSE (iter139), and GECCO3 models are 154 Sv, 140 Sv, 162 Sv, and 141 Sv, respectively (Table 3). For simulated periods for each reanalysis, the ECCOv4r5, ECCO-LLC270, and GECCO3 ACC strengths do not show unrealistic trends showing rather stable values, consistent with observations (Hogg et al., 2015; Donohue et al., 2016). We note, however, that the B-SOSE model's ACC (iter105) increases by ∼30 Sv within 5 years with a maximum strength of 200 Sv (Fig. 9), so B-SOSE (iter105)'s unrealistic ACC strengthening is fixed in the iteration 139. As ACC transport is a

235 good measure for evaluating global simulations, we recommend checking ACC transport when using newer versions of ocean reanalysis products.

The estimates of WG and RG strengths are uncertain due to intense sea-ice cover, lack of observations, and large seasonal and internannual variability. The estimated WG strength has a wide range of values between 30 and 100 Sv based on previous

observation-based and modeling studies (e.g., Fahrbach et al., 1991; Park and Gambéroni, 1995; Beckmann et al., 1999; Wang and Meredith, 2008; Mazloff et al., 2010; Cisewski et al., 2011; Reeve et al., 2019; Neme et al., 2023). The estimated RG strengths are between 10-30 Sv (Chu and Fan, 2007; Mazloff et al., 2010; Nakayama et al., 2014b) and the most recent study using altimetry data estimated the RG strength to be 23±8Sv (Dotto et al., 2018). Following previous studies, we calculate the strengths of the WG and RG by extracting the minimum stream function values in the Weddell Sea and Ross Sea, respectively. In the ocean reanalyses, simulated WG strengths are 56 Sv, 60 Sv, 54 Sv, and 35 Sv for ECCOv4r5, ECCO-LLC270, B-SOSE (iter139), and GECCO3 models, respectively (Table 3). The simulated RG strengths are 21 Sv, 32 Sv, 32 Sv, and 15 Sv for ECCOv4r5, ECCO-LLC270, B-SOSE (iter139), and GECCO3 models, respectively (Table 3). We note that GECCO's WG and RG strengths are weaker by about 30-50%, which may be explained by the fact that ECCOv4r5, ECCO-LLC270, and SOSE use ERA products, while GECCO uses NCEP products as atmospheric forcing as initial guesses. Due to the large uncertainty of observational estimates, mean WG and RG strengths of ocean reanalyses are consistent with observations (Fig. 9).

The ACC strength from the reanalyses shows seasonal and interannual variabilities with magnitudes of about 5 Sv, which is small considering the total ACC strength of about 150 Sv (Figs. 9-11). Seasonally, the ACC tends to be marginally stronger in winter than in summer, a pattern observed consistently across all reanalyses (Fig. 11a). For the interannual variability, simulated small fluctuations are consistent with observations, as Gutierrez-Villanueva et al. (2023) find almost no interannual changes in ACC strengths using repeated oceanographic measurements from 2005 (Fig. 10a).

The WG strength from the reanalyses shows a substantial seasonal and interannual variability (Figs. 9-11). For the interannual variability, time series of negative sea level anomalies in the center of the gyre obtained from SSH observations between 2011 to 2019 suggest an acceleration of the gyre beginning in 2014 and ending in 2017 (Armitage et al., 2018; Auger et al., 2022b), which seems to be captured to some extent in these ocean reanalyses (Fig. 10b). There is, however, a substantial disparity in the interannual variability among the reanalyses (Fig. 10b). For the seasonality, the gyre tends to strengthen during winter and weaken during summer (Fig. 11b), which aligns with SSH observations from 2013 to 2019 (Auger et al., 2022b).

The RG strength demonstrates substantial interannual variability according to reanalyses, though it exhibits less marked seasonal changes than the WG (Figs. 9-11). Observations of the Ross Gyre indicate a slowdown starting in 2015 and continuing into 2016 (Dotto et al., 2018; Armitage et al., 2018). This is replicated across all reanalyses with varying amplitude (Fig. 10c). Notably, observations of the RG display distinct seasonal variability, weakening in summer and intensifying bi-annually in autumn and spring (Dotto et al., 2018; Auger et al., 2022b). While the winter slowdown is consistently observed in all reanalyses, there are differences in how they represent the intensity and timing of the bi-annual intensification (Fig. 11c).

The spatial structures of the ACC, WG, and RG are also similar across all ocean reanalyses (Figs. 4k-n). For example, the simulated 2°C isotherms, which roughly represent the southern extent of the Antarctic Circumpolar Current (ACC), are consistently located in all reanalyses. Additionally, the stream functions within the ACC exhibit similar patterns across the reanalyses. To assess the agreement between the reanalyses and observations, we employ the Mean Dynamic Topography (MDT) as a proxy. Although the MDT only represents the circulation of the surface geostrophic currents, it enables a qualitative comparison of the spatial structure of the Southern Ocean circulation. The MDT derived from the regional dataset (Fig. 4o) including ice-covered measurements from Armitage et al. (2018) demonstrates good agreement with the stream functions

obtained from the reanalyses within the ACC. The WG structures also exhibit similarity across all ocean reanalyses and observations, as indicated by the zero stream function contours extending from 60°W to 120°E. The simulated RG size (defined here using zero stream function contour) is larger in ECCO-LLC270, extending from 160°E to 100°W, while ECCOv4r5 and GECCO3 show smaller RG extending from 160°E to 140°W. While the MDT derived from satellite altimetry confirms a larger RG, it does not account for deeper currents, water column height, or ageostrophic movement.

### 3.4 Weddell and Ross Gyre hydrography

We find qualitatively good agreement between ocean reanalyses and observations for the Southern Ocean. However, polar oceanographers and glaciologists care about modest changes in potential temperature as on-shelf warming by 0.5 or 1°C can substantially enhance ice shelf melting and impact abyssal overturning circulation (e.g., Dutrieux et al., 2014; Li et al., 2023). Here, we further evaluate MITgcm-based ocean reanalyses by focusing on detailed structures and time evolution in the open-water region and comparing them against in-situ observations.

### 3.4.1 Weddell Gyre hydrography: mean state

We use the repeatedly measured oceanographic sections from Joinville Island toward Kapp Norvegia (Section SR04 in Fig. 1a). Based on observations, we find four main water masses: Antarctic Surface Water (ASW), Warm Deep Water (WDW), Weddell Sea Deep Water (WSDW), and Weddell Sea Bottom Water (WSBW) as shown in Figs. 12i-j and 13i-j (e.g., Vernet et al., 2019). For example, in the central part of the WG, the Antarctic Surface Water is found at the top 50-100 m, characterized by a cold (colder than 0°C) and fresh (fresher than 34) layer. The WDW is located above the 0°C isotherms, WSDW is located between 0°C and −0.7°C, and WSBW is located below −0.7°C (Figs. 12-13). At the southwestern edge of the section over the continental shelf, we find a thickened layer of ASW. Over the slope, we find a warm core of WDW (or mCDW) flowing towards the Filchner-Ronne Ice Shelf (see red circles in Figs. 12i-j). At the northwestern edge of this section, the thickening of the ASW over the continental shelf is also observed. Over the continental slope, we find a layer of cold ($< -1$°C) and fresh ($<34.65$) water over the continental slope (blue ellipses in Figs. 12i-j and 13i-j, S3), which is the newly formed Antarctic Bottom Water originating from the Filchner-Ronne Ice Shelf.

For the ocean reanalyses along the same section, all reanalyses show similar hydrographic characteristics. They all show (a) thick ASW over the continental shelf and (b) cores of WDW at the southwestern side of the section (Figs. 12-13). None of the simulations, however, show a layer of cold and freshwater over the continental slope at the northwestern side of this section, because none of the ocean reanalyses resolve the downslope descent of dense shelf water and formation of AABW from the Weddell Sea (Figs. 12-13, S3).

### 3.4.2 Weddell Gyre's hydrography time evolution

For most global ocean simulations without hundreds of years of spin-up, the simulated changes in the deep ocean are mostly caused by model drifts, a phenomenon well-recognized among ocean modelers (Rahmstorf, 1995). As the model's deep ocean

circulation, mixing, diffusion, etc., are different from the real ocean, the model's state gradually shifts from the observed quasi-steady state to the model's steady state. Thus, we need to pay extra attention when analyzing time evolution at depths. When users want to discuss deep ocean changes, it is imperative to verify that (1) the magnitude of simulated changes exceeds that of model drift and (2) the identified changes stem from other external forcing mechanisms. In this section, we demonstrate the influence of model drifts for the Weddell Gyre section by comparing it with repeated observations.

Strass et al. (2020) studied the observed time evolution of deep water characteristics in the Weddell Gyre (WG) and found warming trends at most sampling locations. The mean warming rate is about $\sim$0.002°C per year below 3000 m, which exceeds that of the global ocean by a factor of about 5. We find a similar level of warming by comparing observations in 1989 and 2010 (Table 4). Salinity also increases at most sites below 700 m. This salinity increase is not strong enough to fully compensate for the warming effect on seawater density and deep water masses show a general density decrease as shown in Strass et al. (2020).

Despite bottom water property changes attracting scientific attention, these observed changes in temperature and salinity in the deep Weddell Gyre are small and water mass characteristics are highly stable. For example, in comparing observed sections for 1989, 1995, 2005, and 2010, it is difficult to spot differences for both temperature and salinity (Figs. 12i, 13i). When comparing temperature and salinity for 2010 and 1989, the observed temperature and salinity trends at the center of the WG below 2000 m are about $\sim$0.0019°C per year and $\sim$0.0001 per year, consistent with Strass et al. (2020).

The simulated WG sections show all observed water masses (ASW, WDW, WSDW, and WSBW) for the entire simulated periods for all ocean reanalyses (Figs. 12, 13). Year 1 shows excellent agreement with ocean reanalyses and observations since all the reanalyses are initialized with interpolated ocean observations such as the World Ocean Atlas (Locarnini et al., 2018; Zweng et al., 2019) with some adjustments by adjoint methods. However, all the reanalyses simulate larger changes than the observations. Such changes in deep watermasses can also be seen in the entire deep Southern Ocean as shown in Fig. 8.

For simulated potential temperature, ECCOv4r5 and GECCO3 show warming of WDW (located roughly at 1500-2000 m) and WSBW (located roughly below 3000 m) as shown in Fig. 12 and Table 4. The simulated warming of WDW in these two ocean reanalyses exceeds 0.0084°C per year (three times the observed value), while the observed mean temperature increase between 1500-2000 m is 0.0028 °C per year (Fig. 12 and Table 4). ECCO-LLC270 shows moderate warming but the cores of WDW inflow along the continental slope show a maximum warming rate of 0.01 °C per year. B-SOSE (iterations 105 and 139)

exhibits complex patterns with both warming and cooling and their simulated maximum and minimum trends are much larger than observations (Figs. 12e-f and S4). Thus, in all reanalyses, simulated water mass changes for both WDW and WSBW are much larger than observations.

    The simulated ECCOv4r5 and ECCO-LLC270 salinity increases below the halocline. These rates for both WDW and WSBW are over 10 times larger than observations (Fig. 13 and Table 4). B-SOSE demonstrates slightly different behaviors with

some patches of freshening close to the coast but deep water shows a large salinity increase similar to other ocean reanalyses (Figs. 13, S5, Table 4). GECCO presents the best agreement with the least salinity changes. Since observations show that the temperature, salinity, and thus density of the deep WS water masses remain relatively stable, ocean reanalyses fail to reproduce realistic changes. They all show excessive temperature and/or salinity changes (Table 4).

The Hovmöller diagrams (Fig. 14) show simulated changes in water mass characteristics for the middle of the WG (see blue dot in Fig. 1 for the location). All ocean reanalyses considered in this study show gradual change, which is the model adjustment to the model steady state from the initial condition. Among all the ocean reanalyses considered in this study, ECCO-LLC270 shows the smallest temperature and salinity changes, which are still larger than observations (Table 4). For ECCO-LLC270 temperature, we find simulated cooling above 1500 m and warming deeper than 1500 m. ECCO-LLC270 salinity increases for all depths below the near-surface halocline by approximately 0.02 after 26 years. Other reanalyses show larger trends than that of ECCO-LLC270 (Table 4). In ECCOv4r5, two warming peaks exist at depths of approximately 1000 m and 3000 m, representing the warming of WDW and AABW. B-SOSE simulates warming for all depths, with maximum warming below the mixed layer. For GECCO3, excessive warming can be found at most depths, and the maximum warming of 0.63°C is simulated at a depth of ∼1700 m. The ECCOv4r5, B-SOSE, and GECCO3 salinity show freshening for depths shallower than the halocline, and salinity increases below the halocline.

### 3.4.3  Ross Gyre's hydrography mean state

We utilize the S4P oceanographic section (Fig. 1) that traverses the Ross Gyre and the Pacific sector of the Southern Ocean. We identify three main water masses (Fig. 15): Antarctic Surface Water (ASW), CDW, and AABW. In the western part of the section (corresponding to the Ross Gyre), we observe CDW or modified CDW at shallower depths and AABW at deeper depths. On the other hand, in the eastern part of the section, a warmer core of CDW is observed for the top 2000 m, and we can find AABW below CDW (Fig. 15i). We compare the hydrographic characteristics of the ocean reanalyses along the same section and found that all simulations exhibited similar features (Figs. 15, 16). Similar to the Weddell Gyre section (Figs. 12,13), none of the simulations show a layer of cold and freshwater over the continental slope at the western side of this section, because all MITgcm-based ocean reanalyses fail to simulate the descent of dense shelf water and the formation of AABW.

### 3.4.4  Ross Gyre hydrography time evolution

In this section, we present an additional example where temporal evolutions of the deep ocean are not in good agreement with observations. Specifically, observations from the Ross Sea exhibit more significant warming and freshening than the Weddell Sea. Thus, our example in this section highlights how both model drifts and external forcing influence deep ocean circulation and hydrography. It appears that model drifts predominantly drive the simulated changes, as most simulations consistently show changes that are substantially larger than observed changes.

According to Purkey and Johnson (2013); Purkey et al. (2019), observations of the S4P section suggest the Ross Sea has warmed and freshened below 2000 m at a depth mean rate of 0.002–0.004 °C year$^{-1}$ and ∼0.001 year$^{-1}$, respectively. Along S4P we find smaller changes between 1989 and 2018 (Figs. 15, 16 and Table 4) compared to observed changes (Purkey et al., 2019) because we also include the eastern part of the S4P section for our analyses where water masses present smaller changes.

Here, we plot simulated RG sections for different years (1, 5, 10, and the final year) for each ocean reanalysis (Figs. 15-16). Hovmöller diagrams (Fig. 17) also show simulated changes in water mass characteristics for the middle of the RG (see blue dot Fig. 1 for the location). The simulated sections display three water masses (ASW, CDW, and AABW), consistent with

observations. Again, Year 1 demonstrated excellent agreement with observations similar to the Weddell Sea. For the modified CDW temperature changes (between 1500-2000 m), most ocean reanalysis show larger changes than observations (Fig. 15 and Table 4). Potential temperature changes for B-SOSE (iter139) seem to show the best agreement with observations (Table 4), but this is an artifact of spatial averaging. B-SOSE (iter139) shows strong warming and cooling and spatial patterns do not agree with observations. For the AABW (below 3000 m), most ocean reanalyses show too strong warming. The simulated changes exceed observed changes except for B-SOSE (iter 105). Again, potential temperature changes for B-SOSE (iter105) seem to show the best agreement with observations (Table 4), but this is an artifact of spatial averaging with strong cooling and warming on the western and eastern sides, respectively (not shown). The reason for the simulated excessive warming is likely the model drift and the lack of AABW formation in these ocean reanalyses. For the modified CDW salinity changes, all simulation shows larger changes in magnitude compared to observations. For depths between 1500 and 2000 m, ECCOv4r5, ECCO-LLC270, and B-SOSE (iter139) simulations show a salinity increase, while B-SOSE (iter 105) and GECCO3 show freshening (Fig. 16 and Table 4). Deeper than 3000 m, most ocean reanalyses show increases in salinity, while observations show freshening.

## 3.5 Continental shelf

For the continental shelf regions of Antarctica, a few regions have been attracting attention in the past decades for the changes observed in Antarctic ice sheet loss. There are permanent features, such as mCDW intrusions into the West Antarctic ice shelves and the formation of Dense Shelf Water (DSW) over the Ross and Weddell Sea, that never changed during the period of observations. Thus, we compare last-year slices of simulated on-shelf vertical sections of each ocean reanalysis in the Amundsen Sea, Weddell Sea, Ross Sea, and off the Totten Ice Shelf at the East Antarctic coast (Fig. 1) to check in the first order if models can simulate such permanent features based on observations.

For the Amundsen Sea, we plot repeatedly measured vertical sections connecting open water to the Pine Island Ice Shelf since the first measurements in 1994 (e.g., Hellmer et al., 1998; Jacobs et al., 2011; Nakayama et al., 2013; Dutrieux et al., 2014). For the hydrography in the Amundsen Sea, we find mCDW and WW and past observations show that these two water masses are always present with little change in water mass properties (Figs. 18, 19). Change of the thermocline depth controls the melting of the eastern Amundsen Sea ice shelves Dutrieux et al. (2014); De Rydt et al. (2014). For the ocean reanalyses, ECCOLLC270 and GECCO3 show very shallow thermocline depth likely caused by outdated bathymetric datasets used for these simulations (also seen in Fig. 1), choices of atmospheric forcing, lack of ice shelf cavities, etc. ECCOv4r5 and B-SOSE show better agreement with observations for the Pine Island Trough but still show shallower thermocline depths and warmer mCDW compared to observations.

For the Ross Sea, we compare the vertical section roughly following the Ross Ice Shelf front. This section plotted is a composite of the observations in 1984, 1994, and 2007 (Jacobs and Giulivi, 2010). The observed sections were all made during the summer months. Under warm AASW, the western Ross Sea is filled with high salinity shelf water (HSSW), and the central section is characterized by the presence of MCDW inflow and ice shelf water (ISW) outflow. The modeled sections all suffer from flooding by CDW (note the change in color for temperature sections). The lack of HSSW in all datasets is likely attributed

to the poorly simulated sea ice production rates, lack of ice shelf cavities for LLC270, SOSE, and GECCO, and over-estimated onshore flux of mCDW due to too large GM diffusivity over the continental slope (Dettling et al., 2023). Lastly, even though ECCOV4r5 has an open cavity, the ISW with observed values cannot be produced, as the appropriate source water mass is missing. None of the reanalysis reflects the freshening trend (e.g., Fig 5 in Jacobs and Giulivi, 2010).

For the Weddell Sea, we compare the vertical section roughly following the Filchner-Ronne ice shelf front. This section has been repeatedly observed many times and we select 1989, 1996, 2005, and 2010 for model-data comparison. The modeled sections are much closer to observations than in the Ross Sea, in temperature, salinity, and bathymetry, with the exception of B-SOSE. ECCO-LLC270 and GECCO3 show a modified Warm Deep Water (mWDW) core in the Western part of the section but do not have HSSW or ISW in the Filchner trough. ECCOv4r5 shows the opposite section structure: it is able to produce the

colder water mass in the trough (ISW in case of ECCOv4r5 in the presence of open cavity) but does not show mWDW inflow (Janout et al., 2021). BSOSE shows a thick fresh layer at the top of the water column and a warm intrusion below it, similar to the Ross Sea.

     For East Antarctica, we select a recently observed section off the Totten Ice Shelf. The first observational campaign was conducted in 2015 (Rintoul et al., 2016) and repeated a few times after that (Hirano et al., 2023; Nakayama et al., 2023).

The section is constructed in the following way: left-side panels are along-front and right-side panels are away toward the continental shelf break (Figs. 18, 19, fourth column). Based on observations (Figs. 18t,19t), we find thick winter water (WW) and a thin layer of mCDW close to the bottom. In contrast to the Amundsen Sea, thick WW regulates both the volume and temperature of mCDW flowing into the Totten Ice Shelf cavity (Hirano et al., 2023). All reanalyses show too thick mCDW, and mCDW temperature is more than 1°C warmer than observations at the Totten Ice Shelf front. ECCO-LLC270 and GECCO3

show mCDW filling the continental shelf up to 200m.

     We note that BSOSE iteration 139 differs significantly from iteration 105 close to the coast (Fig. S6). While the later iteration removed a cold and fresh water column immediately adjacent to the front of PIG, the significant freshening of the top 200 m in the Ross and Weddell Seas has likely stopped DSW and HSSW production in both regions and led to the flooding of the continental shelves by CDW.

## 430   **4   Discussion**

### **4.1   Downscaling regional simulation for the Southern Ocean**

In recent regional simulations of the Southern Ocean, ocean reanalyses have been used to construct lateral boundary conditions. For instance, Nakayama et al. (2017, 2018, 2021a) utilized ECCO-LLC270 to simulate West Antarctic and East Antarctic configurations of the regional model. Other researchers (e.g., Ito, 2022) used B-SOSE to construct their model boundary

condition and study carbon cycles in the Central Pacific Sector of the Southern Ocean.

     As discussed in the previous section, all MITgcm-based ocean reanalyses simulate large-scale Southern Ocean hydrography and circulation patterns similar to observations in the open ocean, especially in the Weddell Sea. None of the models, however, accurately capture the time evolution of the Weddell and Ross Gyre hydrography, which contains the most stable (not changing)

and repeatedly measured watermasses in the Southern Ocean. All ocean reanalyses show excessive warming trends. Moreover, all reanalyses fail to successfully simulate on-shelf hydrographic structures and circulations. Simulated on-shelf conditions are different from observations (e.g., different water masses found on the shelf, missing water masses, etc).

Therefore, when using ocean reanalyses to force regional ocean simulation, we recommend that (1) northern regional model boundaries should be located away from the continental shelf in the area with enough observations and the east-west extent of the model should be large enough to minimize the effect of boundary conditions on the study region and (2) intensive model-data comparison should be repeated to evaluate the new model performance. Sometimes, we can locate the east and west model boundaries at locations where we find little inflow into the model domain (e.g., Amundsen-Bellingshausen domain (Nakayama et al., 2017, 2018). Then, we can minimize the influence of on-shelf model boundary conditions (ocean reanalyses on-shelf hydrographic properties). For example, by doing so, the Amundsen Sea and East Antarctic regional simulations (Nakayama et al., 2018) successfully capture interannual changes of on-shelf ocean heat intrusions. We caution that the choice of boundary condition is crucial for studying near-surface processes due to the large spread of surface properties amongst ocean reanalyses (Fig. 5).

## 4.2 Common problems and possible solutions

There are several common problems in the simulation of Southern Ocean states that may be addressed through various solutions.

One major issue is that all ocean reanalyses struggle to accurately simulate the formation of Antarctic Bottom Water (AABW) and its precursor Dense Shelf Water and exhibit excessive warming of deep water (Figs. 12,15). One potential solution is to implement a bottom boundary layer scheme that can control the transport of dense shelf water to deep layers. Studies using coarse-resolution NOAA/GFDL simulations have shown that implementing such a scheme can significantly improve the representation of AABW formation and deep water masses (Snow et al., 2015). Increasing horizontal and vertical resolution can also help address this problem (Mensah et al., 2021), but this may only be practical for regional simulations due to the massive CPU time required for global simulations with fine grid spacing (e.g., Stewart et al., 2019). Conducting data assimilation requires an additional 5-10 times more CPU hours/computational time. Modifying the weight for cost calculation is another possible solution, as the current version of ocean reanalyses has too few observations in the Southern Ocean and is weakly constrained. Applying larger weights for stable water masses based on existing observations or separating the cost into climatological mean and anomaly components can help put different weights on the mean state and time evolution, which has been implemented for ECCOv4r5 and for the last 11 iterations for GECCO3. An alternative solution could be to conduct data assimilation for shorter time windows (e.g., a few months) as is done for the California Current ocean state estimates (Zaba et al., 2018). This allows us to increase model controllability while still enforcing governing physics and capturing the continuous evolution of large-scale dynamics over weeks to months. However, it also means that the state estimates can not provide closed budgets over longer timescales as discontinuities will exist between each assimilation window.

Another common problem is that seasonal and interannual variabilities of large-scale ocean circulation, such as Weddell Gyre (WG), Ross Gyre (RG), and Antarctic Circumpolar Current (ACC) strengths, do not agree among ocean reanalyses,

despite using sea surface height data to constrain their models (Figs. 9-11). This issue can be addressed by implementing newly emerging datasets for the polar region, such as surface height anomaly datasets that include sea ice-covered regions (Armitage et al., 2018; Dotto et al., 2018; Auger et al., 2022a). These datasets can help optimize seasonal and interannual changes in sea surface height and thus barotropic ocean circulation changes.

Next, fresh bias in the Sea Surface Salinity (SSS) in the Amundsen, Bellinsghausen and Ross Seas in reanalyses needs to be taken into account when creating a regional model boundary condition that crosses the continental shelf (Fig. 5). The fresh water is confined to the Coastal Current circling the continent, thus the eastern boundary of the regional domain will import the fresh water into the domain. The regional domain may experience a warm SST bias due to the shallow mixed layer.

Lastly, none of the ocean reanalyses successfully simulate on-shelf hydrography close to observations (Figs. 18-19). Nakayama et al. (2021b) demonstrated that adjoint optimization can improve the representation of on-shelf hydrography if there are enough observations to constrain the model. Oceanographic observations are accumulating, particularly over the Antarctic shelf region close to rapidly melting ice shelves, and can help constrain ocean models. Mooring observations at critical locations (e.g., Webber et al., 2017; Darelius et al., 2023) and massive datasets from ship-based and seal-tag CTD measurements can be particularly useful. More deployment of under-ice capability Argo floats are underway and new satellite-based datasets are emerging, such as the temporally varying Antarctic ice shelf melt rate estimates Adusumilli et al. (2020); Paolo et al. (2022). These newly emerging datasets can further help constrain the ocean reanalyses for polar oceans.

## 5 Conclusions

We assess the performance of MITgcm-based ocean reanalyses for the Southern Ocean. For the mean states, the reanalyses agree well with sea ice, observed hydrographic structures, ocean circulation in the open ocean (e.g., Antarctic Circumpolar Current and Weddell and Ross Gyre circulations), and spatial distributions of MLD.

For the time evolution, MITgcm-based ocean reanalyses show good agreement with observations for sea ice concentration for both seasonal cycles and interannual variability. Sea ice volume is also consistent with observations despite the large errors of satellite-based estimates. However, MITgcm-based ocean reanalyses do not always agree with observations. We need to pay extra attention and we recommend that each user conduct their own evaluation to support their research goals. For example, most reanalyses successfully reproduce sea ice seasonal and interannual variability. However, there is no coherent time evolution of the ACC, WG, and RG strength, indicating a need for improvements in the representation of sea surface height and barotropic stream function for the Southern Ocean region. We also find excessive trends in the Weddell and Ross Gyres' deep water hydrography, which are concerning given their observed temperature and salinity do not change significantly over time.

For the continental shelf region, we do not recommend using global MITgcm-based ocean state estimates for both mean states and time evolution. All reanalyses fail to reproduce the observed state. For most regions, we find substantial differences (e.g., different water masses found on the shelf or missing water masses), which suggests that different physical processes govern ocean circulation and determine hydrographic structures. This is a concern for recent studies (Walker and Gardner,

2017; Bronselaer et al., 2018; Rignot et al., 2019; Brancato et al., 2020; Millan et al., 2020; Kim et al., 2021) that utilize ocean reanalyses for studying ocean melting ice shelves, as they may not yet have the skill to produce meaningful output. Development of downscaling simulation of MITgcm-based ocean reanalyses is a solution for continental shelf region and we can achieve much model-data agreement including both mean states and time evolution (Nakayama et al., 2017; Taewook et al., 510 2024).

Overall, while the MITgcm-based reanalyses generally perform well globally, there is room for improvement in capturing variability in the open ocean and mean states over the Antarctic continental shelves.

*Code and data availability.* ECCOv4r5 data is available from https://zenodo.org/records/10930853 (Nakayama, 2024a) or https://ecco.jpl. nasa.gov/drive/files/Version4/Release5. ECCO-LLC270 data is available from https://zenodo.org/records/10934678 (Nakayama, 2024b) and 515 https://zenodo.org/records/10935131 (Nakayama, 2024c) or https://ecco.jpl.nasa.gov/drive/files/Version5/Alpha. SOSE iterations 105 and 139 are available from https://zenodo.org/records/10935396 (Nakayama, 2024d) and https://zenodo.org/records/13117970 (Nakayama, 2024e), respectively, or from http://sose.ucsd.edu/. GECCO3 is available from https://www.fdr.uni-hamburg.de/record/14187 (Köhl, 2024) or https: //www.cen.uni-hamburg.de/en/icdc/data/ocean/reanalysis-ocean/gecco3.html. Analysis scripts developed for this manuscript and observational data can be obtained from https://zenodo.org/records/13270783 (Nakayama and Malyarenko, 2024).

*Author contributions.* Y.N. conceived the study. Y.N. and A.M. conducted the model and data analyses and wrote the initial draft of the paper. Y.N., A.M., M.A., H.Z., O.W., Y.N., I.F., M. M., A. K., and D.M. discussed the results and implications and commented on the manuscript at all stages.

*Competing interests.* The authors declare no competing interests.

# 6 Acknowledgments

This work was also supported by the fund from Grant in Aids for Scientific Research (21K13989, 24K15256, and 24H02341) of the Japanese Ministry of Education, Culture, Sports, Science, and Technology. This work was also supported by the Inoue Science Research Award from the Inoue Science Foundation and JSPS Bilateral Joint Research Projects between Japan and Germany (JPJSBP-120233501). The research was also carried out at the Jet Propulsion Laboratory, California Institute of Technology, under a contract with the National Aeronautics and Space Administration (NASA). Support was provided by 530 an appointment to the NASA Postdoctoral Program; the NASA Cryosphere program; and the NASA Modeling, Analysis, and Prediction program. AM is supported by the New Zealand Ministry of Business, Innovation and Employment Antarctic Science Platform (grant no. ANTA1801). MA is supported by the Australian Research Council Special Research Initiative, Australian Centre for Excellence in Antarctic Science (Project Number SR200100008). AK acknowledges support from the EARTH$^{RA}$

project funded by the Deutsche Forschungs Gemeinschaft through Reinhart Koselleck projects. MRM acknowledges support from NSF awards OCE-1924388, OPP-2319829, OPP2149501, and OPP-1936222, and NASA grants 80NSSC24K0243, 80NSSC20K1076, and 80NSSC22K0387.

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

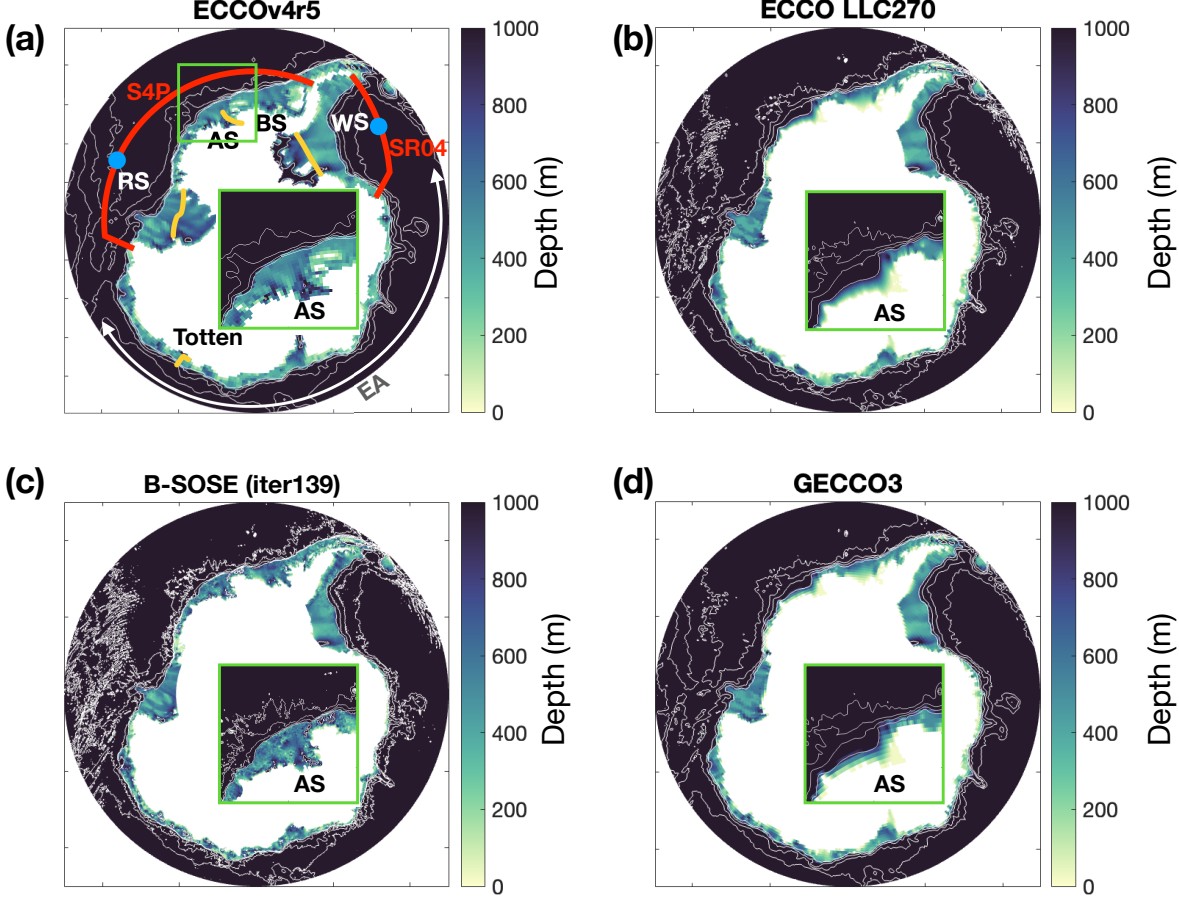

**Figure 1.** Model bathymetry (color) of (a) ECCOv4r5, (b) ECCO-LLC270, (c) B-SOSE, and (d) GECCO3 with contours of 1000 m, 2000 m, 3000 m, and 4000 m in white. Letters RS, AS, BS, WS, and EA denote the Ross Sea, Amundsen Sea, Bellingshausen Sea, Weddell Sea, and East Antarctica, respectively. For all panels, insets show close-ups of the Amundsen region enclosed by green boxes. Red lines show the locations of vertical sections SR04 and S4P. Orange lines show the location of vertical on-shelf sections shown in Figs. 18 and 19. Blue dots indicate the location where Hovmöller diagrams are shown in Figs. 14 and 17.

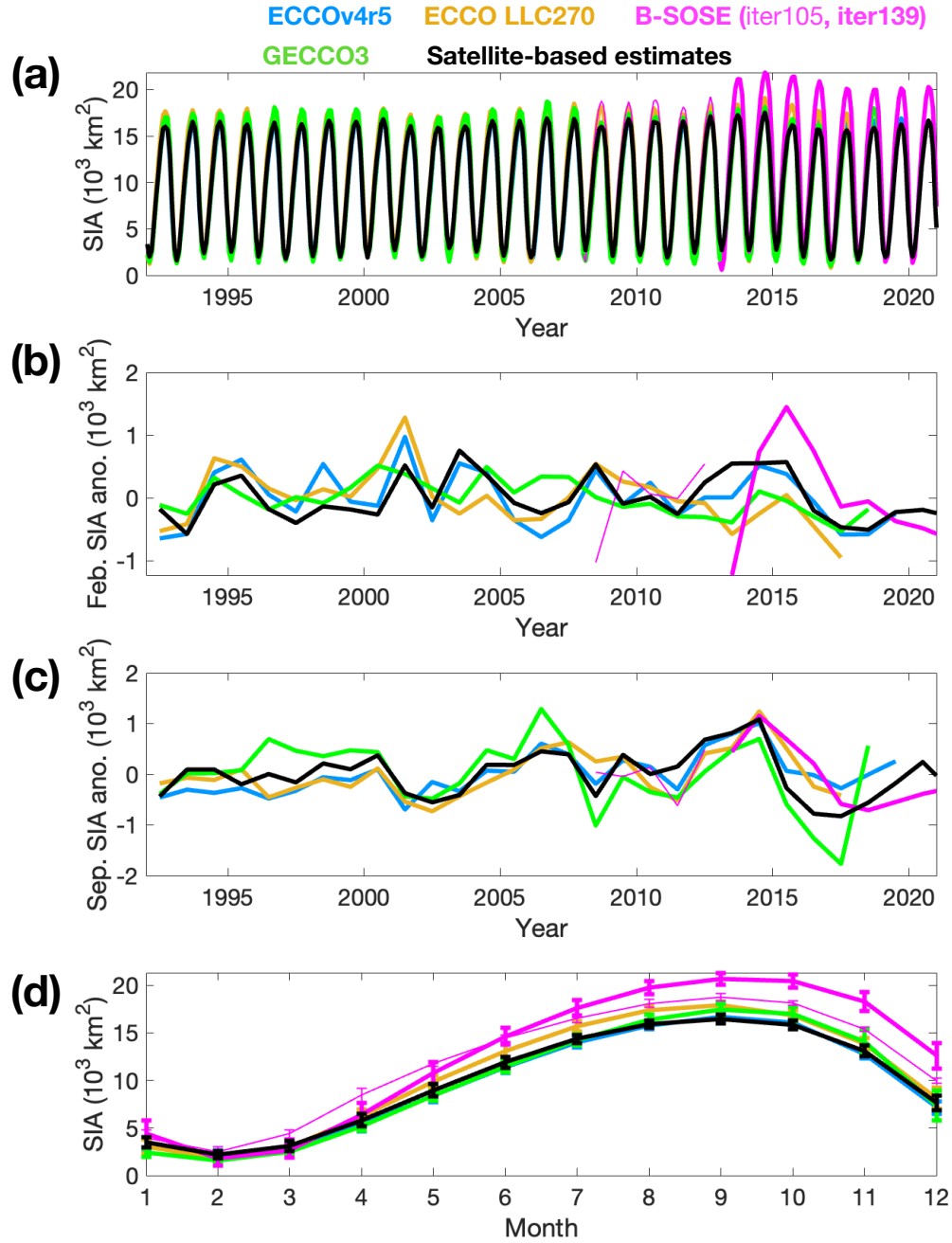

**Figure 2.** (a) Time series of sea ice area (SIA), (b, c) SIA anomalies for February and September, and (d) climatological SIA seasonal cycle. ECCOv4r5, ECCO-LLC270, B-SOSE (iter105), B-SOSE (iter139) GECCO3, and satellite-based estimates are shown in blue, orange, magenta (thin), magenta, green, and black, respectively. For (d), error bars represent one standard deviation of variability. The 5-day SOSE outputs are averaged monthly for comparison with other datasets.

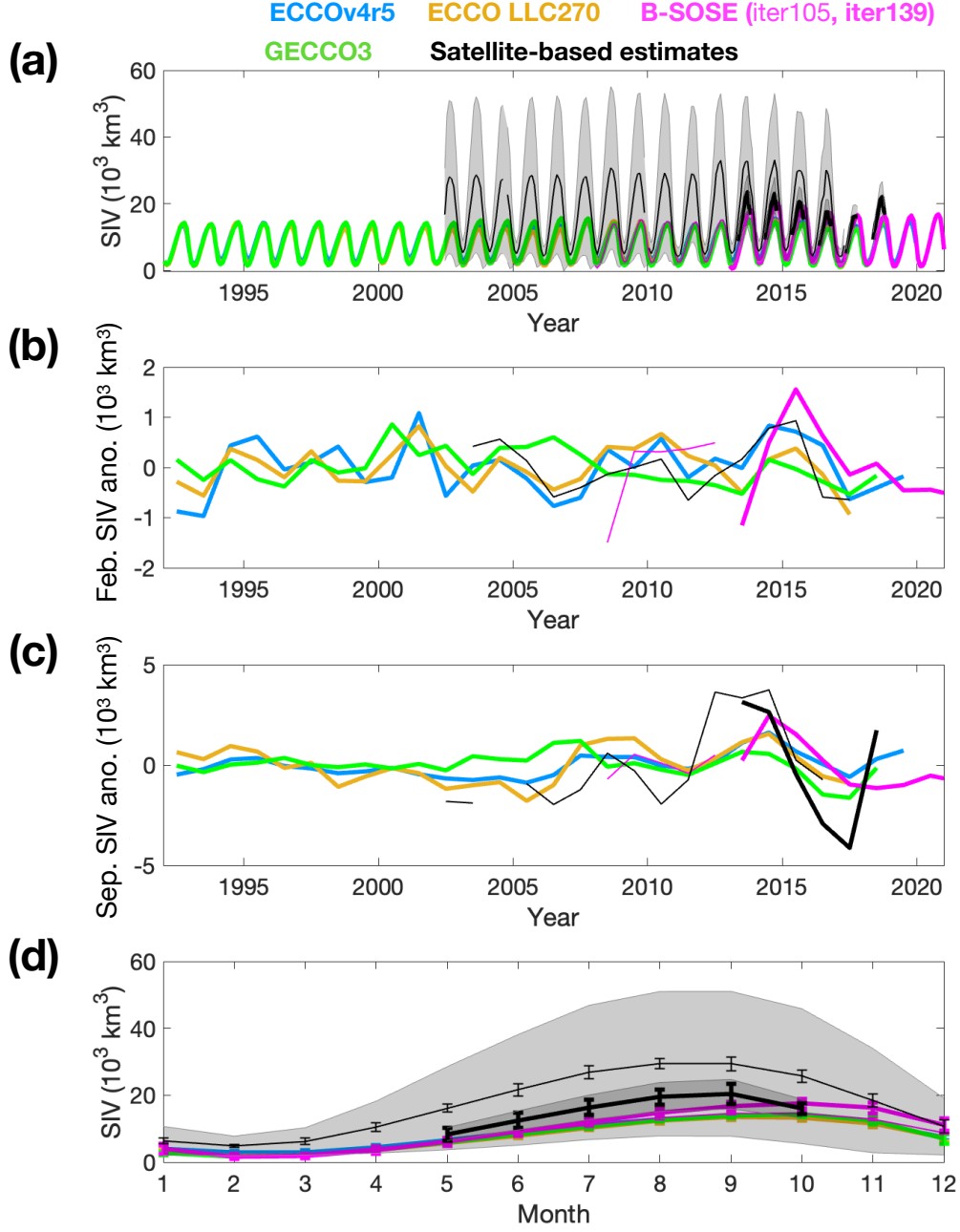

**Figure 3.** Same as Figure. 2 but for sea ice volume (SIV). Two observational data are plotted, thin and thick black lines indicate products from SICCI and LEGOS, respectively. The shadings in (a) and (d) represent the uncertainties of the observations.

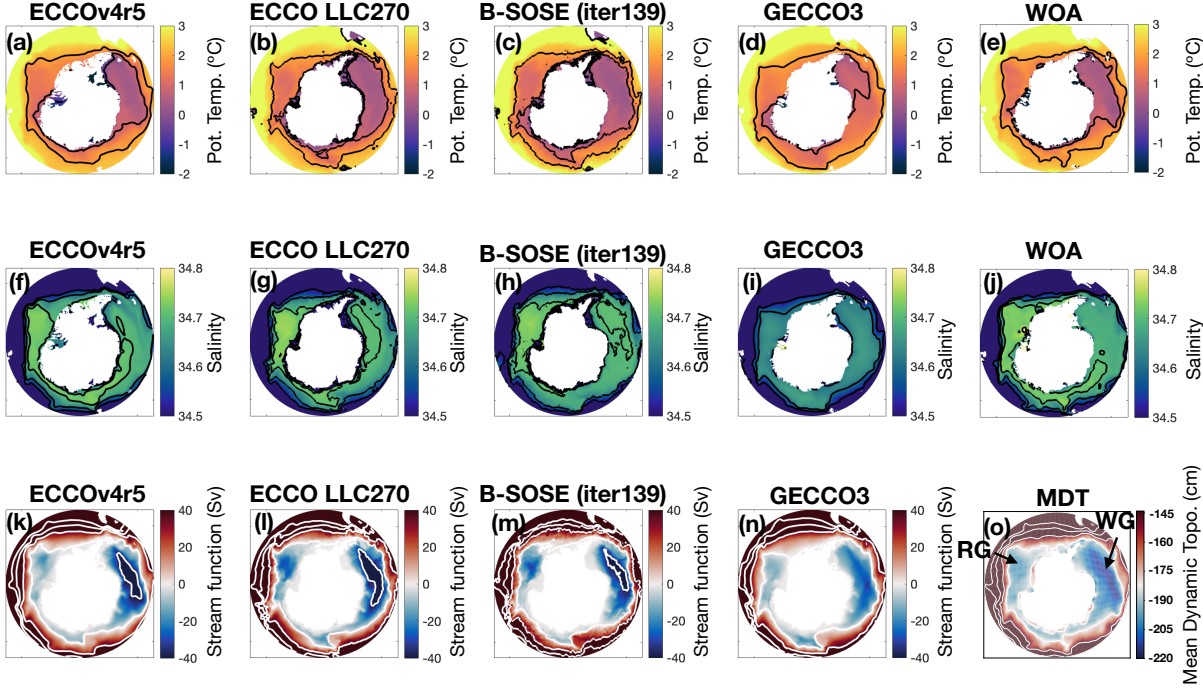

**Figure 4.** (a,f,k) ECCOv4r5, (b,g,l) ECCO-LLC270, (c,h,m) B-SOSE, (d,i,n) GECCO3 mean potential temperature and salinity, and stream function for the model simulated periods, respectively. (e,j) World Ocean Atlas climatology potential temperature and salinity, respectively. (o) Mean dynamic topography from Armitage et al. (2018). WG and RG denote the Weddell Gyre and Ross Gyre, respectively. For (a)-(e), potential temperature contours of -1 °C, 0 °C, 1 °C, and 2 °C are shown. For (f)-(j), salinity contours of 34.5, 34.6, 34.7, and 34.8 are shown. For (k)-(o), stream function contours of -40 Sv, 0 Sv, 40 Sv, 80 Sv, and 120 Sv are shown. As the vertical grids are slightly different, we extract potential temperature and salinity at 553 m, 553 m, 552 m, 560 m, and 550 m for ECCOv4r5, ECCO-LLC270, B-SOSE, GECCO3, and WOA, respectively.

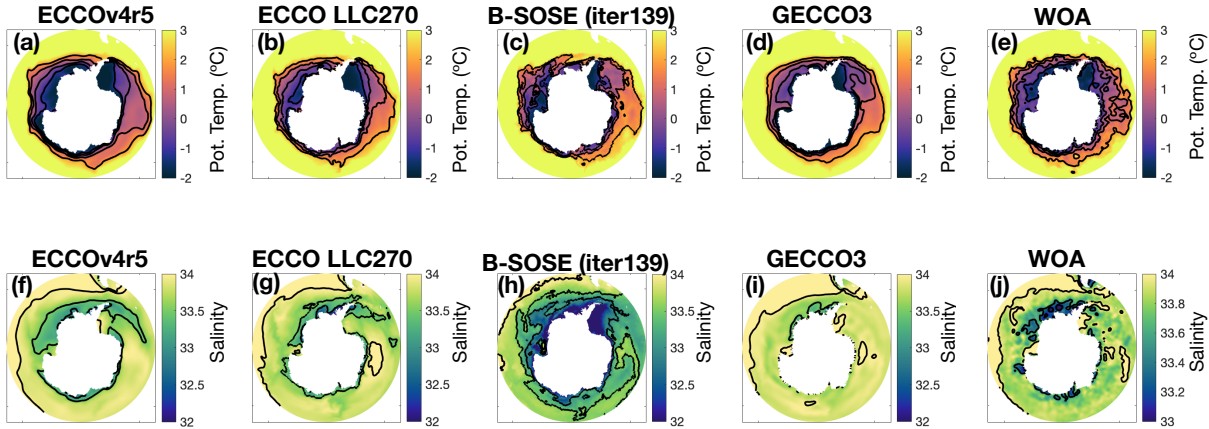

**Figure 5.** (a,f) ECCOv4r5, (b,g) ECCO-LLC270, (c,h) B-SOSE, (d,i) GECCO3 February mean surface potential temperature and salinity. (e,j) World Ocean Atlas (WOA) climatology potential temperature and salinity, respectively. For (a)-(e), potential temperature contours of -1 °C, 0 °C, 1 °C, and 2 °C are shown. For (f)-(j), salinity contours of 33.0, 33.5, and 34.0 are shown. Note that the color scale for WOA salinity (j) is different from that of the other figures.

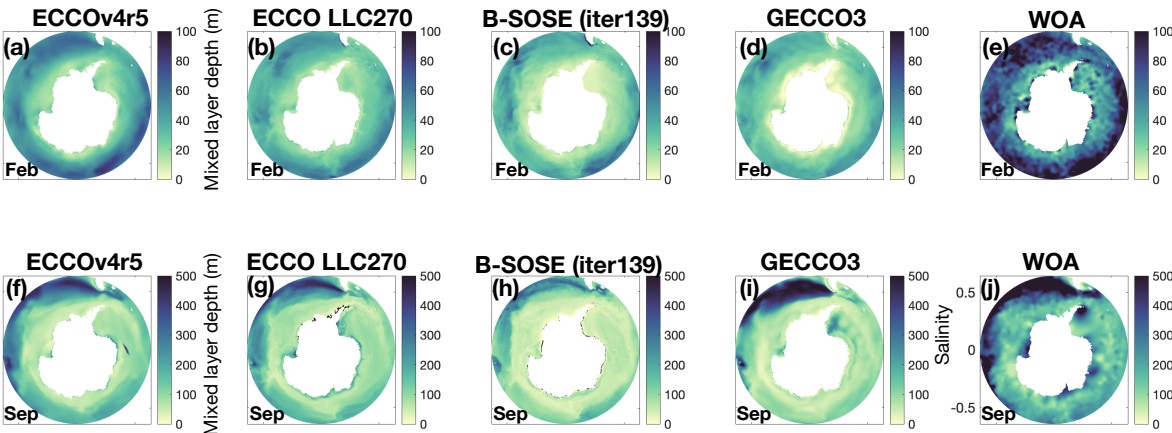

**Figure 6.** (a,f) ECCOv4r5, (b,g) ECCO-LLC270, (c,h) B-SOSE, (d,i) GECCO3 February and September mean MLD. MLD is defined as the depth level where the density difference from the surface exceeds 0.03 kg m$^{-1}$. (e,j) World Ocean Atlas monthly climatology of MLD for February and September, respectively.

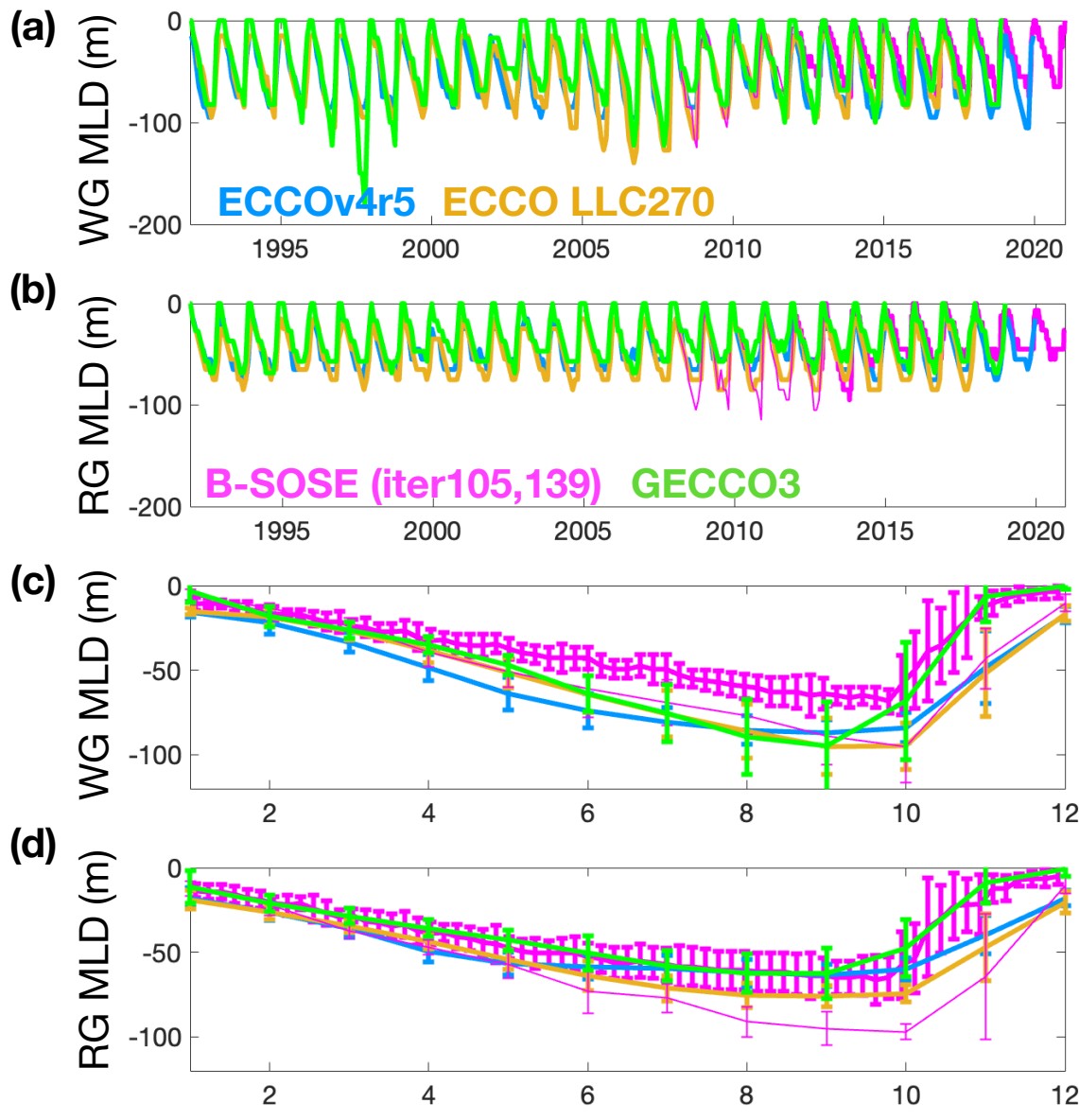

**Figure 7.** Time series of MLD for the (a) Weddell Gyre and (b) Ross Gyre for locations shown by blue dots in Figure 1. Climatological MLD seasonal cycle for (c) Weddell Gyre and (d) Ross Gyre. For (c,d), error bars represent one standard deviation of variability. ECCOv4r5, ECCO-LLC270, B-SOSE (iter105), B-SOSE (iter139), and GECCO3 are shown in blue, orange, magenta (thin), magenta and green, respectively.

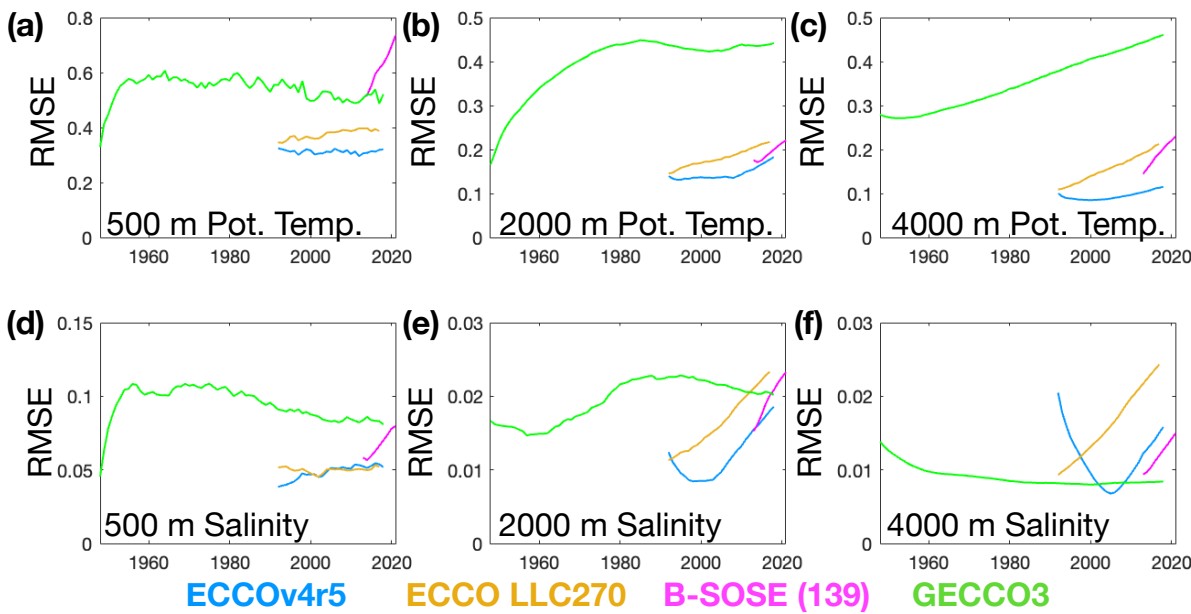

**Figure 8.** Time series of RMSE of (a,b,c) potential temperature and (d,e,f) salinity at 500 m, 2000m, and 4000 m depth levels, respectively. RMSE is calculated by interpolating each ocean state estimate output into a common horizontal grid with 1° resolutions. ECCOv4r5, ECCO-LLC270, B-SOSE (iter105), B-SOSE (iter139), and GECCO3 are shown in blue, orange, magenta (thin), magenta and green, respectively.

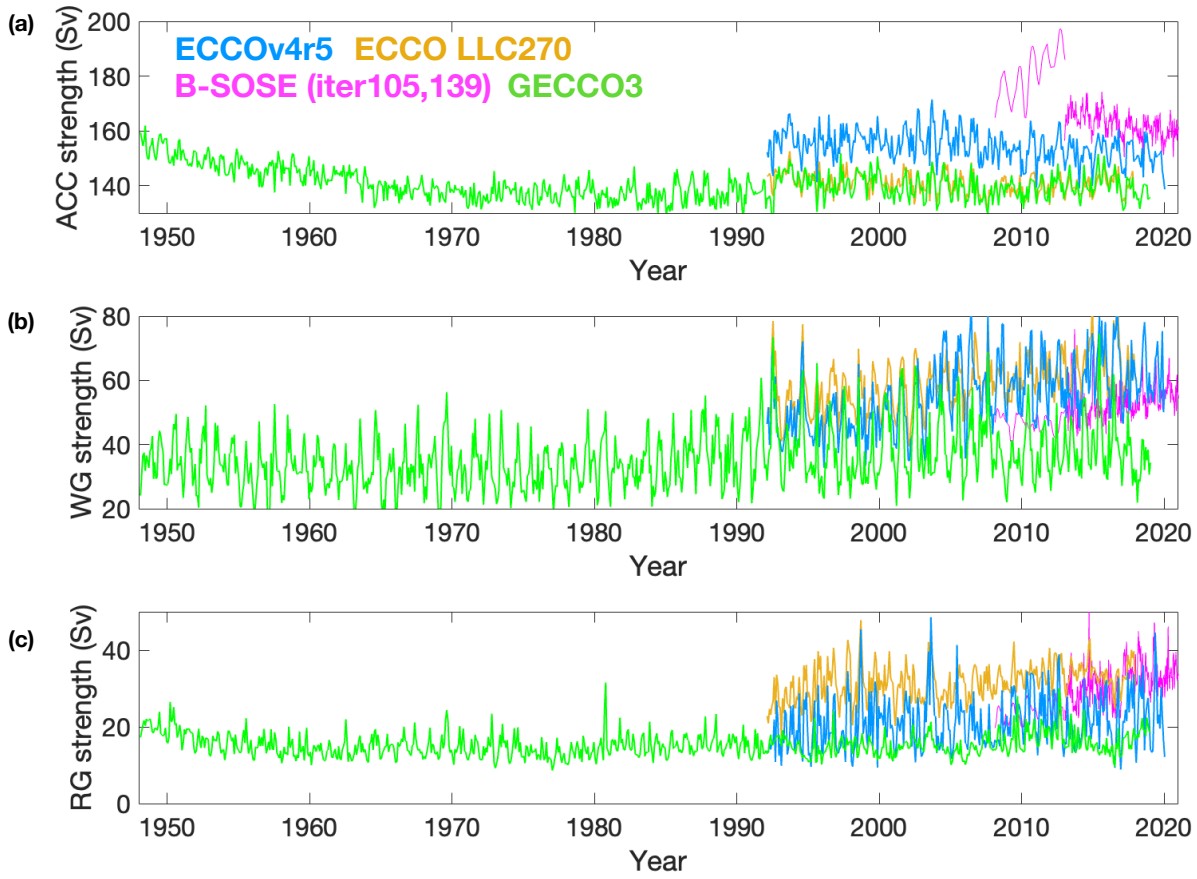

**Figure 9.** Time series of the strength of (a) ACC, (b) WG, and (c) RG stream function. ECCOv4r5, ECCO-LLC270, B-SOSE (iter105), B-SOSE (iter139), and GECCO3 are shown in blue, orange, magenta (thin), magenta and green, respectively.

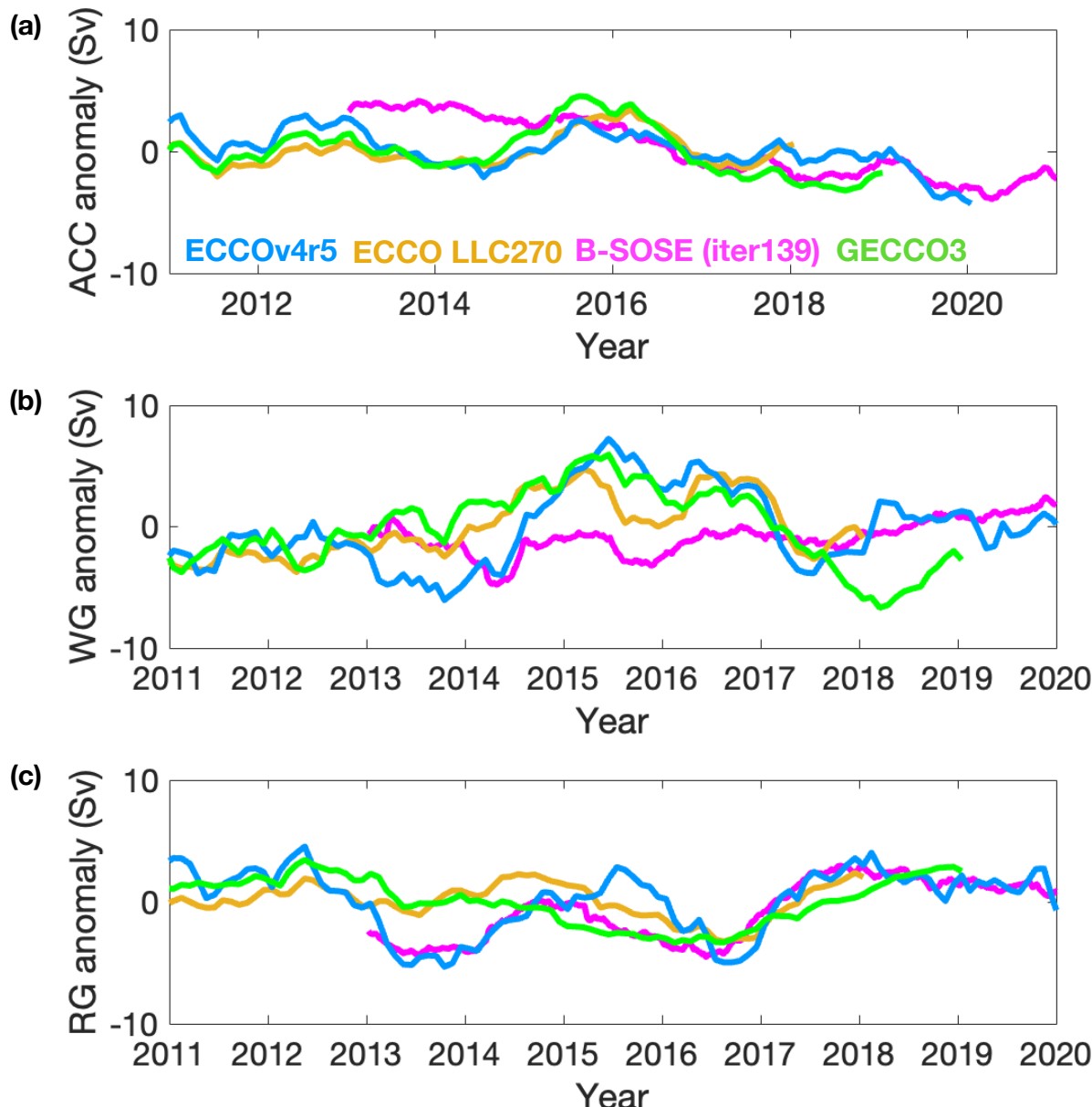

**Figure 10.** Same as Fig. 9 but for ACC, WG, and RG anomalies.

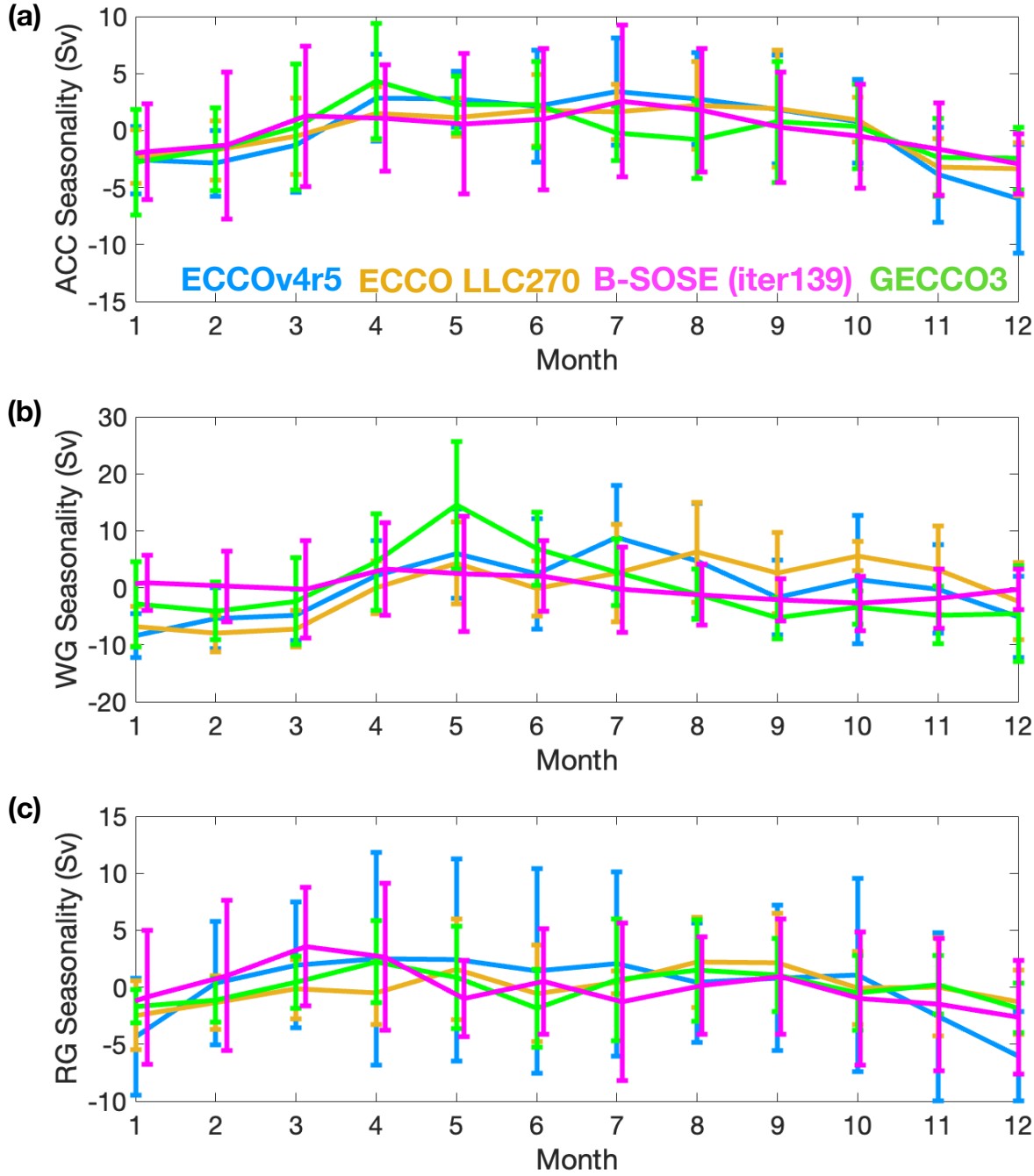

**Figure 11.** Same as Fig. 9 but for ACC, WG, and RG seasonality. Error bars represent one standard deviation of variability.

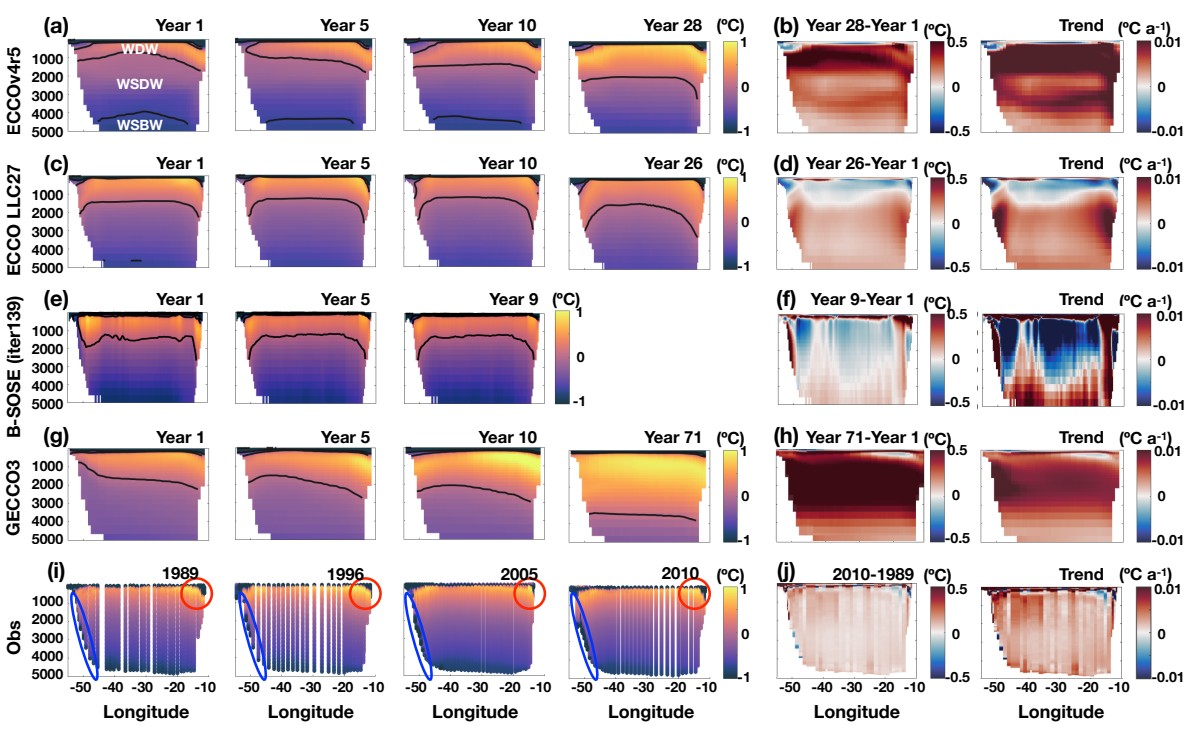

**Figure 12.** Simulated SR04 vertical sections of (a) ECCOv4r5, (b) ECCO-LLC270, (c) B-SOSE, and (d) GECCO3 potential temperature for year 1, year 5, year 10, and final years of each reanalysis along SR04 section (Fig. 1). B-SOSE only extends for 9 years and only years 1, 5, and 9 are plotted. (i) Observed vertical section of potential temperature for 1989, 1996, 2005, and 2010 along the SR04 section. Potential temperature (b,d,f,h,j) differences and trends (defined as the difference of the first and last years divided by the total model period). The potential temperature contours of -0.7 °C and 0 °C are also in black for panels (a), (c), (e), and (g). Close-ups of the shelf break regions enclosed by blue circles are also shown in Fig. S3.

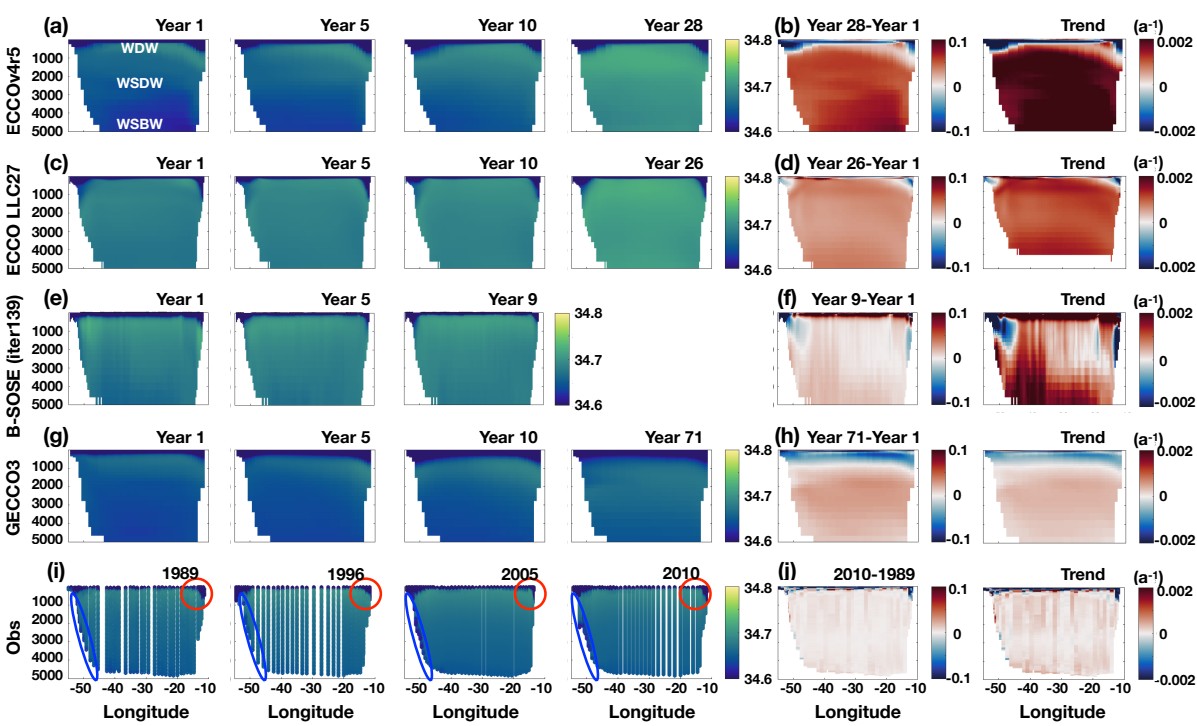

**Figure 13.** Same as Fig. 12 but for salinity. Close-ups of the shelf break regions enclosed by blue circles are also shown in Fig. S3.

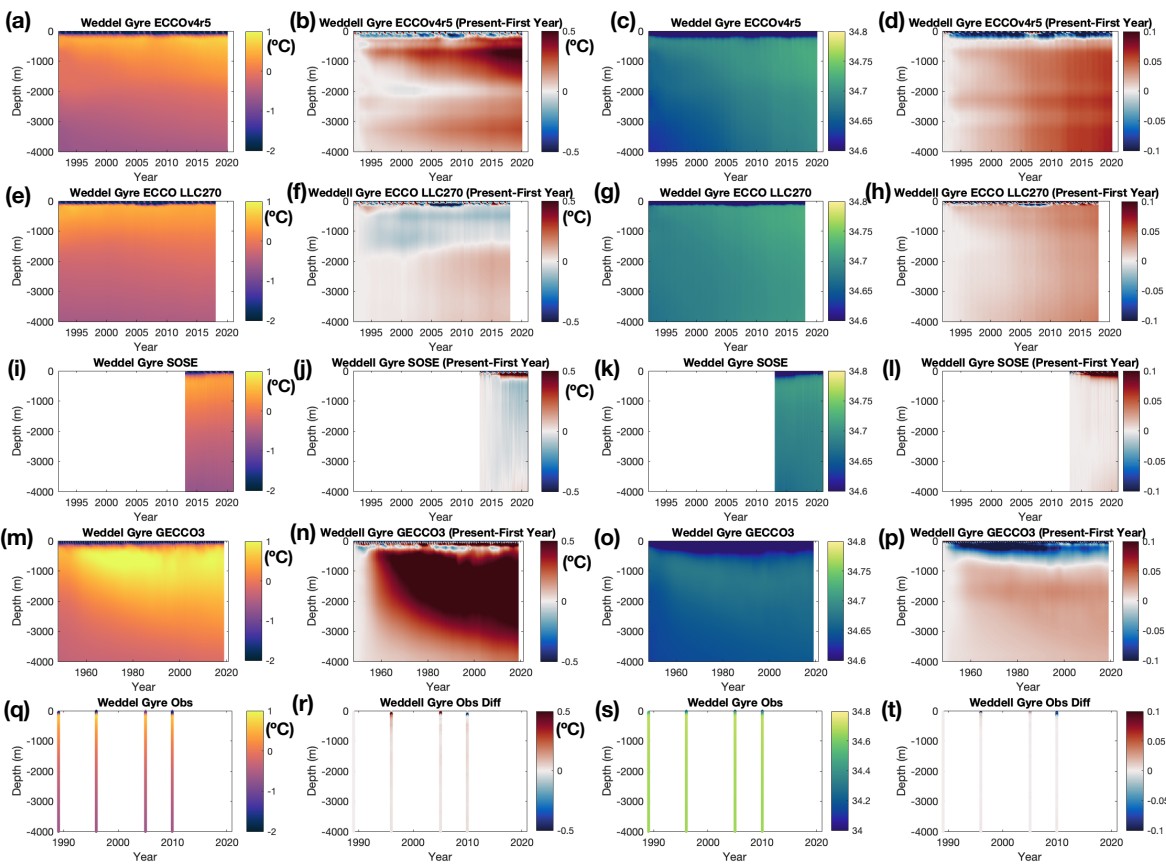

**Figure 14.** Times series of Weddell Gyre (a,e,i,m) temperature and (c,g,k,o) salinity (blue dot in Fig. 1) for ECCO-v4r5, ECCO-LLC270, B-SOSE, and GECCO3. Time series of (b,f,j,n)temperature and (d,h,l,p)salinity difference to year 1 mean for ECCO-v4r5, ECCO-LLC270, B-SOSE, and GECCO3. Vertical plots of observed (q) potential temperature, (r) potential temperature difference to 1989, (s) salinity, and (t) salinity difference to 1989 are also shown. These hydrographic properties are sampled from the nearest locations from $330.0°$E and $66.0°$S.

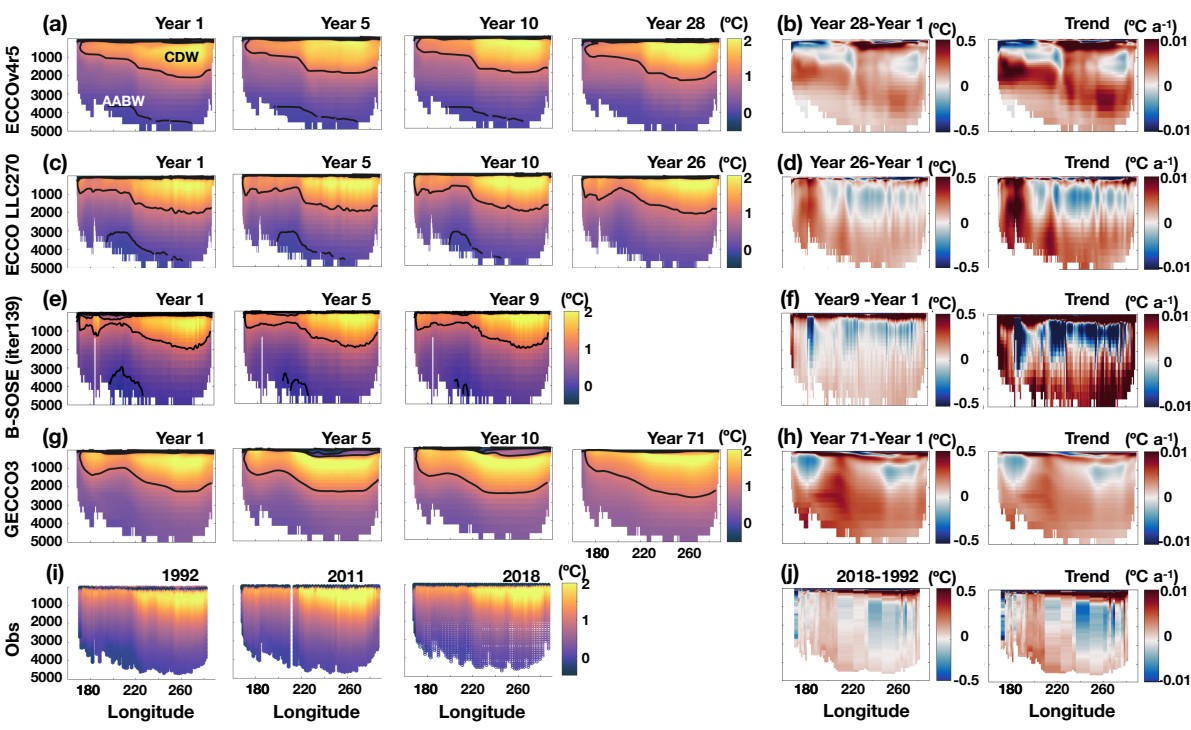

**Figure 15.** Same as Figure 12 but for the S4P section (Fig. 1). The potential temperature contours of 0 °C and 1 °C are also in black for panels (a), (c), (e), and (g).

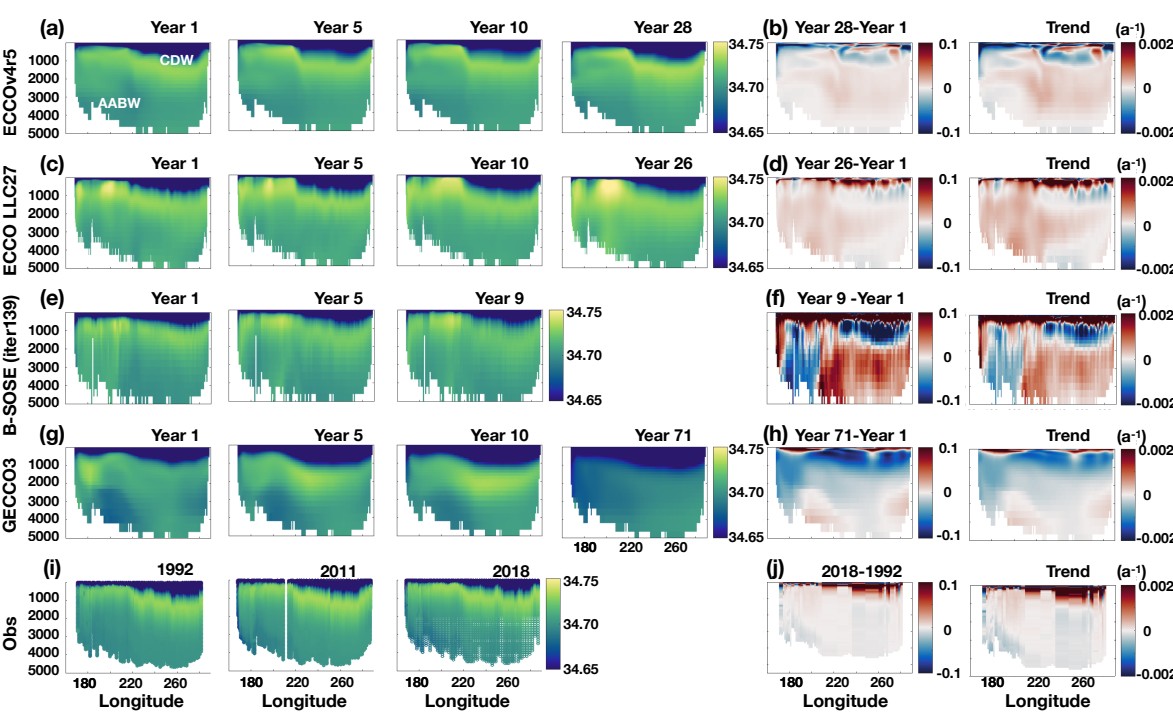

**Figure 16.** Same as Figure 12 but for salinity along the S4P section (Fig. 1).

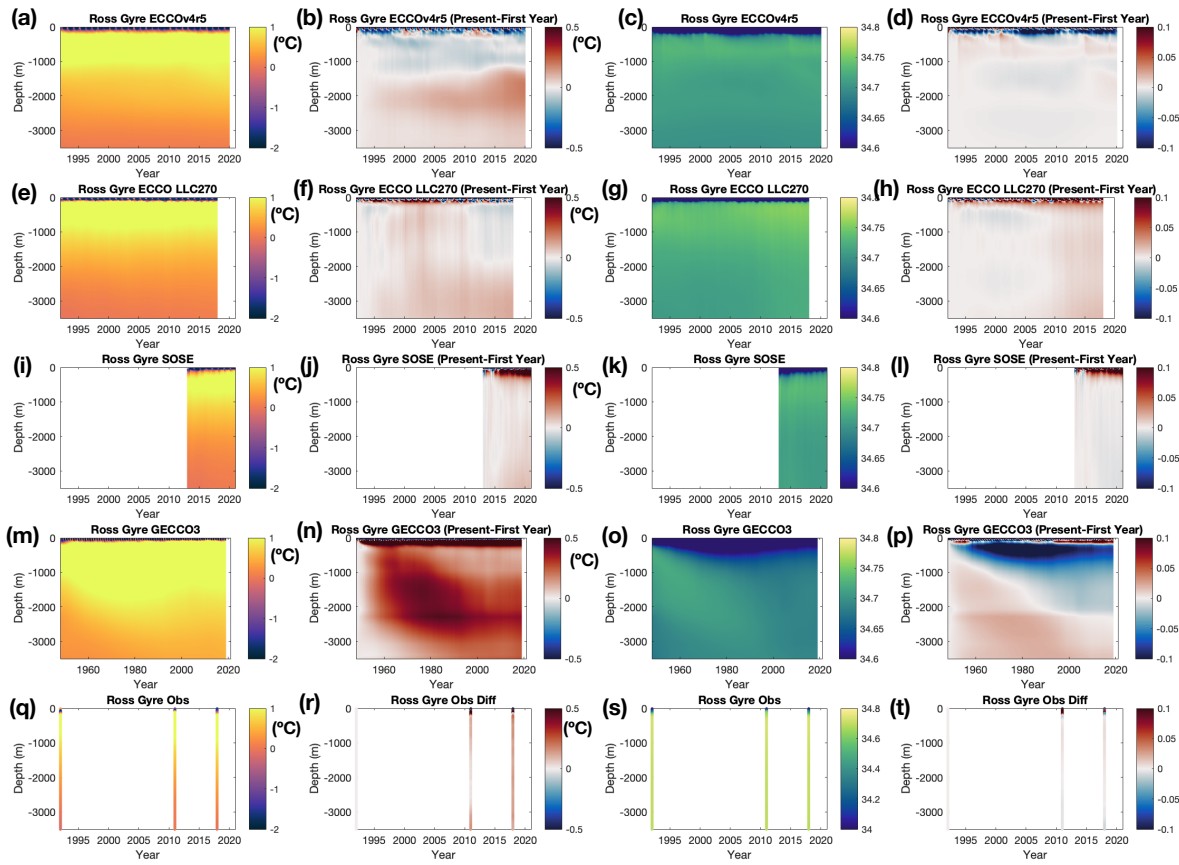

**Figure 17.** Times series of Ross Gyre (a,e,i,m) temperature and (c,g,k,o) salinity (cyan dot in Fig. 1) for ECCO-v4r5, ECCO-LLC270, B-SOSE, and GECCO3. Time series of (b,f,j,n) temperature and (d,h,l,p) salinity difference to year 1 mean for ECCO-v4r5, ECCO-LLC270, B-SOSE, and GECCO3. Vertical plots of observed (q) potential temperature, (r) potential temperature difference to 1992, (s) salinity, and (t) salinity difference to 1992 are also shown. These hydrographic properties are sampled from the nearest locations from 199.7°E and 67.8°S.

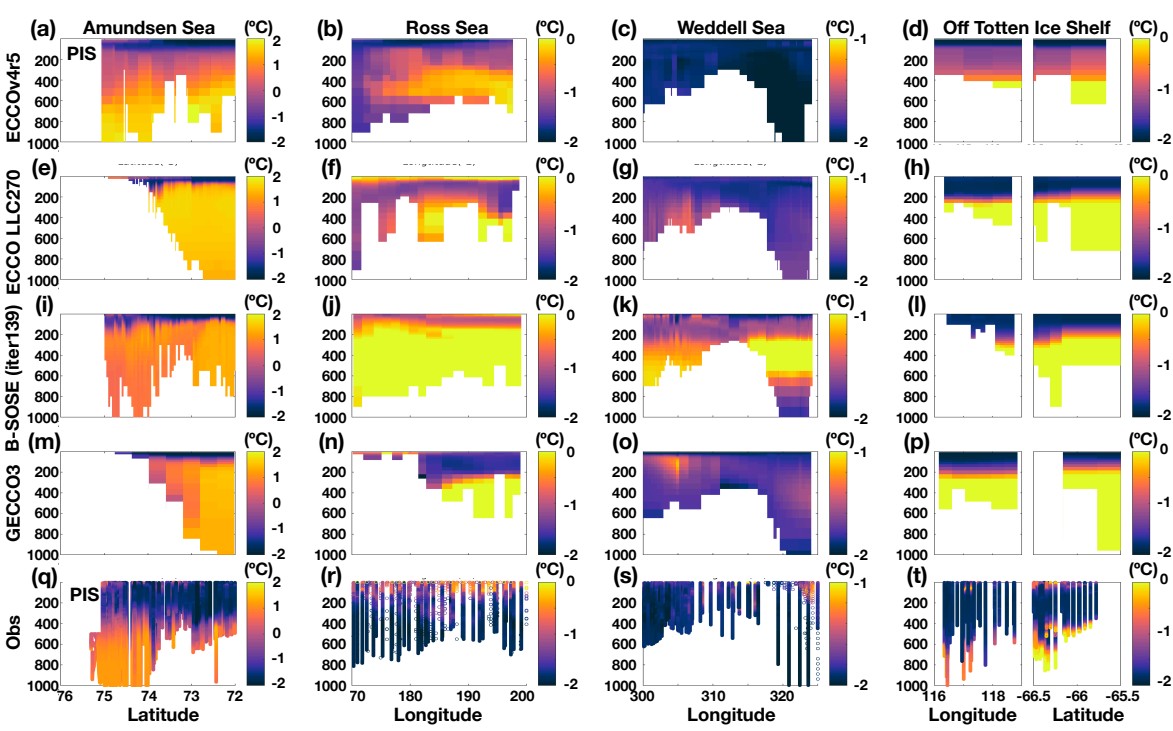

**Figure 18.** Simulated on-shelf vertical sections of (a-d) ECCOv4r5, (e-h) ECCO-LLC270, (j-l) B-SOSE, and (m-p) GECCO3 potential temperature of the final model year for the Amundsen Sea, Ross Sea, Weddell Sea, and Off the Totten Ice Shelf. (q-t) The same observed vertical sections of potential temperature compiled all available years of observations.

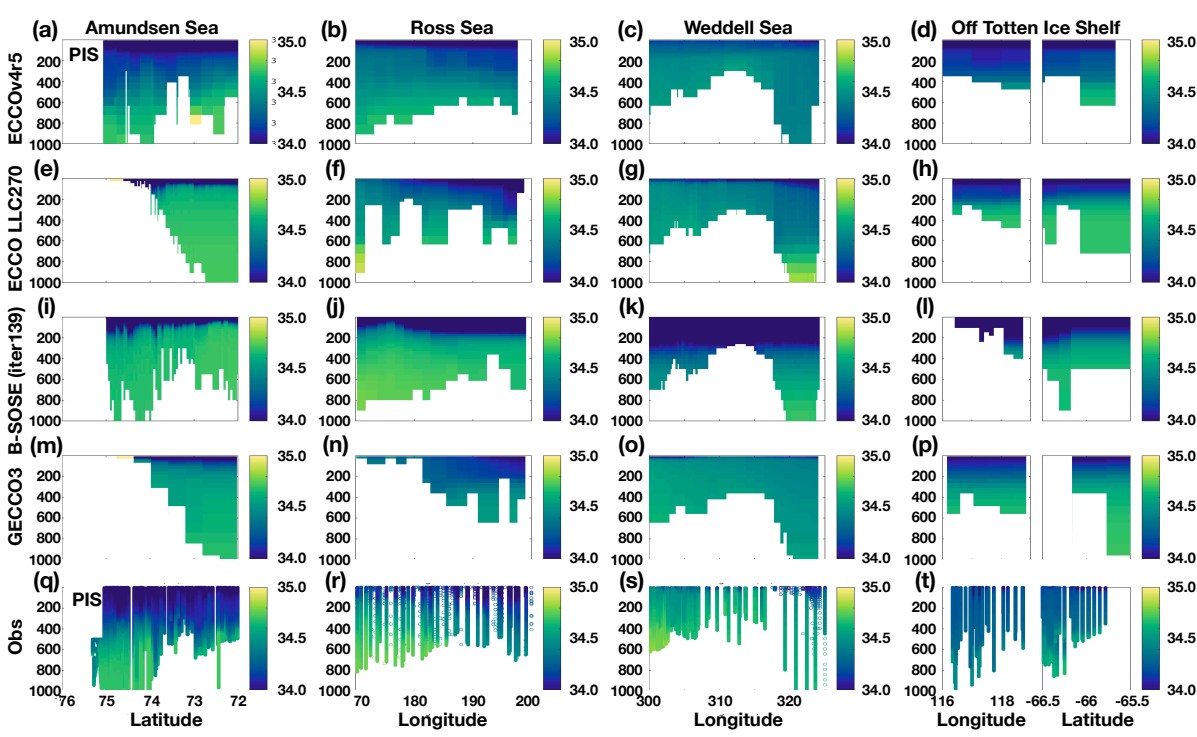

**Figure 19.** Same as Fig.18 but for salinity.

**Table 1.** General information on ocean reanalyses evaluated in this manuscript.

| Name | Year | Horizontal Grid | Reference | Initial atmospheric forcing guess | Bathymetry | Details |
|---|---|---|---|---|---|---|
| ECCOv4r5 | 1992-2019 | $\sim 1°$ | Forget et al. (2015) | hourly MERRA2 | Arndt et al. (2013) | Ice shelf cavities |
| ECCO-LLC270 | 1992-2017 | $\sim 1/3°$ | Zhang et al. (2018) | 6 hourly ERA-interim | Smith and Sandwell (1997) | No ice shelf cavities |
| B-SOSE (iter139) | 2013-2021 | $1/6°$ | Verdy et al. (2017) | Hourly ERA5 | Amante and Eakins (2009) | No ice shelf cavities |
| B-SOSE (iter105) | 2008-2012 | $1/3°$ | Verdy et al. (2017) | 6 hourly ERA-interim | Amante and Eakins (2009) | No ice shelf cavities |
| GECCO3 | 1948-2018 | $\sim 0.4°$ | Köhl (2020) | 6 hourly NCEP/NCAR Reanalysis | Jungclaus et al. (2013) | No ice shelf cavities and surface salinity restoring |

**Table 2.** The standard deviation (SD) of reanalyzed and observed (in parenthesis) SIA anomalies for February, September, and all months. Correlation coefficients followed by **\*\*** indicate that they pass the 95% significance test. The last column represents the root mean square error (RMSE) of the monthly climatology for the 12 months of reanalysis and observation.

| | February | | September | | All months | |
|---|---|---|---|---|---|---|
| | $SD_{SIA}(10^3 \text{ km}^2)$ | Correlation | $SD_{SIA}(10^3 \text{ km}^2)$ | Correlation | $SD_{SIA}(10^3 \text{ km}^2)$ | RMSE of the seasonal cycle ($10^3$ km$^2$) |
| ECCOv4r5 | 0.44 (0.38) | 0.80 ** | 0.40 (0.47) | 0.72 ** | 0.55 (0.55) | 0.36 |
| ECCO-LLC270 | 0,46 (0.37) | 0.34 | 0.45 (0.47) | 0.70 ** | 0.65 (0.53) | 0.90 |
| B-SOSE (iter139) | 0.83 (0.45) | 0.29 | 0.65 (0.67) | 0.63 | 0.99 (0.78) | 3.37 |
| B-SOSE (iter105) | 0.62 (0.31) | -0.59 | 0.40 (0.42) | 0.31 | 0.37 (0.40) | 1.87 |
| GECCO3 | 0.27 (0.38) | 0.20 | 0.67 (0.47) | 0.68 ** | 0.76 (0.54) | 0.74 |

**Table 3.** Simulated mean strengths and standard deviations of the Antarctic Circumpolar Current, Weddell Gyre, and Ross Gyre.

| | ACC | WG | RG |
|---|---|---|---|
| ECCOv4r5 | 154Sv ± 5.6 | 56Sv ± 10.6 | 21Sv ± 7.0 |
| ECCO-LLC270 | 140Sv ± 3.9 | 60Sv ± 8.1 | 32Sv ± 4.8 |
| B-SOSE (iter139) | 162Sv ± 5.3 | 54Sv ± 6.6 | 32Sv ± 6.0 |
| B-SOSE (iter105) | 181Sv ± 9.5 | 48Sv ± 4.1 | 24Sv ± 3.5 |
| GECCO3 | 141Sv ± 5.7 | 35Sv ± 8.9 | 15Sv ± 3.0 |

**Table 4.** Observed and simulated trends in the Weddell Sea and Ross Sea for depth ranges between 1500-2000m and 3000m-bottom. Trends that are three times larger/smaller than the observed values are in bold. Trends in opposite directions are also in bold. Note that simulated trends are calculated for the entire simulated years, which are different for each reanalysis product. Observed trends are calculated by using the temperature and salinity difference for 1989-2010 and 1992-2018 for the Weddell and Ross Sea, respectively.

| | ECCOv4r5 | ECCO-LLC270 | B-SOSE (iter139) | B-SOSE (iter105) | GECCO3 | Observations |
|---|---|---|---|---|---|---|
| WS Pot. Temp. 1500-2000m (°C year$^{-1}$) | **0.0097** | 0.0033 | **-0.0026** | **-0.0060** | **0.0089** | 0.0028 |
| WS Pot. Temp. 3000m-bottom (°C year$^{-1}$) | **0.0061** | 0.0039 | 0.0040 | **-0.0054** | 0.0041 | 0.0021 |
| WS Salinity 1500-2000m (year$^{-1}$) | **0.0020** | **0.0011** | $3.6\times10^{-4}$ | **0.0020** | $3.4\times10^{-4}$ | $1.4\times10^{-4}$ |
| WS Salinity 3000m-bottom (year$^{-1}$) | **0.0023** | **0.0012** | **0.0013** | 0.0012 | $2.5\times10^{-4}$ | $1.2\times10^{-4}$ |
| RS Pot. Temp. 1500-2000m (°C year$^{-1}$) | **0.0045** | **0.0014** | $-1.3\times10^{-4}$ | **-0.016** | **0.0019** | $-7.1\times10^{-4}$ |
| RS Pot. Temp. 3000m-bottom (°C year$^{-1}$) | 0.0034 | 0.0046 | **0.0073** | 0.0014 | 0.0031 | 0.0017 |
| RS Salinity 1500-2000m (year$^{-1}$) | $\mathbf{1.3\times10^{-4}}$ | $\mathbf{1.8\times10^{-4}}$ | $\mathbf{1.1\times10^{-4}}$ | $\mathbf{-1.9\times10^{-4}}$ | $\mathbf{-3.1\times10^{-4}}$ | $-2.2\times10^{-5}$ |
| RS Salinity 3000m-bottom (year$^{-1}$) | $\mathbf{7.1\times10^{-5}}$ | $\mathbf{1.4\times10^{-4}}$ | $\mathbf{2.4\times10^{-4}}$ | $\mathbf{2.0\times10^{-4}}$ | $-3.5\times10^{-6}$ | $-1.0\times10^{-4}$ |