# Peer review of "Evaluation of MITgcm-based ocean reanalyses for the Southern Ocean"

_EGUsphere, 2024_

## Author Comment (AC2)

Author response to the Referee Comments by Anonymous Reviewer 1 on the manuscript:

**Evaluation of MITgcm-based ocean reanalyses for the Southern Ocean**

Yoshihiro Nakayama, Alena Malyarenko, Hong Zhang, Ou Wang, Matthis Auger, Yafei Nie, Ian Fenty, Matthew Mazloff, Köhl Armin, and Dimitris Menemenlis

submitted to Geoscientific Model Developments (https://doi.org/10.5194/egusphere-2024-727)

We thank the Reviewer for the constructive comments. The reviewer's comments are displayed in *italics*, replies are shown in **bold text**.
* * *
*General comments In this study, the authors analyze five different MITgcm general circulation model based ocean reanalyses (3 global and 2 regional) to see how well they simulate the Southern Ocean. Comparisons are made of the total sea ice extent and volume, planar views of the average temperature and salinity at 550m and February mean SST and SSS, time series of ACC and Weddell and Ross Gyre transport, vertical sections and time series of temperature and salinity that match observations from two WOCE sections that are primarily off the continental shelf, and vertical sections of temperature and salinity across four specific sections of the Antarctic continental shelf. The authors find that the different reanalyses "agree well" (line 428) with the observed hydrography and circulation in the open ocean, but generally do not represent observed trends in deep water hydrography and do a poor job of representing hydrography on the Antarctic continental shelf. As a potential user of these products in the Southern Ocean, I'm very happy the authors decided to do this study and I think this will be (at least, selfishly, for me) a fantastic addition to the literature. I certainly think this study is worth the attention of GMD because of how useful it could be for the community. The manuscript was mostly (see below) clear and easy to understand. However, I do have several suggestions that I think will help make this study more suited as a guide for potential future users of these products in the Southern Ocean*

**Thank you very much for your insightful comments.**

*1) There really are not any quantitative comparisons. I didn't see any computations of RMSE, correlations, skill metrics, Taylor diagrams, etc. I realize that quantitative comparisons can be painful to create with different grids and will not be appropriate for every comparison. However, they would be very useful for trying to judge what particular reanalysis gets closest to observations for a specific measurement that might be of most interest to a potential future user (e.g. RMSE for the S4P line for someone who is interesting in the ocean off West Antarctica) beyond just trying to guess from a visual inspection.*

**As suggested, we have included new sea ice area and thickness analyses, including correlation calculations (Figures 2-3 and Table 2), and (b) RMSE analysis mainly for open water. We did not include RMSE for vertical sections because available data is limited and RMSE depends clearly on model year (Fig. 8).**

*2) Some potential users of these reanalyses are not just physical oceanographers and I think a very useful comparison for those interested in biogeochemical uses (although physical oceanographers will be interested as well) would be mixed layer depth characteristics.*

**New analyses of MLD are included in the revised manuscript (Lines 196-211 and Figs 6-7).**

*3) The plan views at one depth and the two WOCE cross section are quite useful for studying the open ocean time-mean hydrography, but are they enough to show that the hydrography in the reanalyses "are consistent with observations" (lines 5-6)? I know it's impractical to look at the plan views for several different depths (and the authors have already presented two depths), but maybe some integrated measures can be displayed. For example, a planar view can be compared of the vertically integrated temperature (or heat content) and salinity (or salt content) over the top 1500 m to that from WOA.*

**We have calculated the RMSE of potential temperature and salinity for 3 different depth levels. Observed water masses in these depth levels show weak seasonal variability and can be used to evaluate the model performance (Fig 8 and Lines 212-221 in the revised manuscript). We did not calculate integrated temperature and salinity because we think RMSE for different depth levels are more direct comparisons. We also agree with reviewers that our results may not be enough to show models' consistency with observations. Therefore, we replace "consistent with observations" with "similar to observations" in some locations in the manuscript.**

*I have several other specific comments and suggestions below, but most of these are minor and should be easily dealt with by the authors. I think this exercise is a great idea, but I do think it needs some work before it will be fully useful for the intended audience.*

*Specific comments*

*Line 18: I think it would be helpful to label the ACC and (especially) the Ross and Weddell Gyres somewhere. I don't know if the best way is to add a schematic as an inset somewhere, label them on one of the stream function or MDT plots, or add a new plot (maybe in the Supplementary Information?).*

**Done. We have added the labels to Figure 4.**

*Lines 33-34: Would it be useful to list some reasons why a regional model (e.g. higher resolution, tuning for a specific region) might be able to match observations better?*

**Done.**

*Lines 38-48: I think the previous study of Uotila et al. (Climate Dynamics, 2019) assessing several ocean reanalyses (including GECCO2) in the polar regions should be mentioned somewhere (maybe not necessarily in this paragraph, but somewhere in the manuscript).*

**Thank you for pointing this out. Uotila et al., 2019 is now mentioned in Line 42.**

*Lines 64-67: I do not agree with this statement as the Uotila et al. study did do a cross-comparison of several reanalyses (although none of the ones here, unless you count the older version of GECCO) and observations for the Southern Ocean.*

**We replace "these products" with "these MITgcm-based ocean reanalysis products" for clarification.**

*Lines 115-122: No details for the ice observations used in section 3.2 are given here (or anywhere else that I could find).*

**Information on the sea ice observations is included as in Lines 119-128.**

*Lines 125-128: The differences in the bathymetry over the continental shelf are an important point (as pointed out in section 3.5 by the authors), but it is very difficult to see this in Figure 1. I suggest an additional figure (here or in the Supplementary Information) with a magnified view of the bathymetry that shows more of the continental shelf differences (perhaps a blow-up of the Amundsen/Bellingshausen sector?).*

**Done (Fig. 1).**

*Line 135: Again, I did not see any information on the sea ice "observations" used in this section.*

**Information on the sea ice observations is included as in Lines 119-128.**

*Lines 140-141: Have the authors looked at the correlation between the modeled and observed September sea ice extents?*

**Correations between simulated and observed sea ice extent are included in Table 2.**

*Lines 147-152: Are there any observed estimates of sea ice volume used here? Even though there is a black "Satellite-based estimates" legend at the top of Figure 3, I don't see any black lines plotted in the figure.*

**Observed estimates of sea ice volume are included in the revised manuscript (Figure 3).**

*Lines 155-156: I suggest adding a little more information here about exactly where 550 m is a good depth for examining CDW intrusions to the Antarctic ice shelves. It certainly is a good depth in the Amundsen-Bellingshausen and (I think) Totten, but it is too deep in some areas (i.e. Ross Sea, Prydz Bay) to show much of open ocean originated MCDW heat heading towards the ice shelves.*

**The manuscript is revised as in Lines 173-175.**

*Lines 166-167: I disagree that the "cold and relatively salty surface water mass" in the Weddell Sea is a "feature of all" estimates: The SSS in the Weddell in B-SOSE (139… the one shown in Fig. 5) is pretty fresh compared to WOA (and all the other reanalyses).*

**The manuscript is revised as in Line 185.**

*Lines 191-194: The ACC transport is based on the flux through Drake Passage, but exactly where is the difference computed in the stream functions to get at the model Weddell and Ross Gyre strengths (apologies if I missed it)?*

**Method is described in Lines 239-240.**

*Line 202: Is there a reference for the statement that SSH observations suggest an acceleration of the WG over 2014-2017?*

**References and additional information on the statement (based on time series of gyre depth) have been added in the sentence Lines 252–257 .**

*Lines 243-246: It is hard to see the cold and fresh layer over the slope at the northwest side of the section in the observations. Is it worth adding a supplemental figure showing a zoomed-in western section?*

**As suggested, close-ups of the western section are included in the supplementary material (Fig. S3).**

*Lines 287-288: Do the authors know of any good references on model drift that could be added here?*

**Rahmstorf, 1995 is cited in the revised manuscript.**

*Lines 299-301: Similar to the comment above for lines 243-246, it is hard to see the cold and fresh layer over the slope at the western side of the section in the observations.*

**As suggested, close-ups of the western section are included in the supplementary material (Fig. S3).**

*Lines 334-336: Do the authors think there might be other reasons, besides the outdated bathymetry, for the shallow thermoclines (i.e. issues with the atmospheric forcing, how glacial ice melt is added, etc.)?*

**We revise the text as in Lines 393-397.**

*Line 343: Is it worth actually computing and comparing the sea ice production rates for the different reanalyses? There are observational based estimates of coastal polynya production and it might be instructive to see how the reanalyses compare.*

**Variables required to calculate sea ice production are not included in the standard output of the MITgcm-based ocean state estimates. Thus, it is outside of the scope of this manuscript.**

*Lines 392-407: What about the lack of HSSW and the poorly simulated sea ice production mentioned earlier in lines 342-343?*

**Revised (Lines 452-453).**

*Technical corrections*

*Line 139: Since this sentence is just about the maximum extent, should "2b-c" be "2b"?*

**Done.**

*Line 140: I think "in March" should be added after "simulated model extent", since the sea ice extent model error is greater than $10^{12} m^2$ in September (Figure 2b).*

**As we made major changes related to including new sea ice volume data, this sentence is removed in the revised manuscript.**

*Lines 143-144: Since this sentence is just about the minimum extent, should "2b-c" be "2c"?*

**Done.**

*Line 165: I did not see any isohalines in Fig. 5.*

**Isohalines are included in the revised manuscript.**

*Line 181 (as well as lines 192, 193, and Table 2): I think it would be helpful if the authors gave the standard deviation for all the model transport estimates, especially since ranges or standard deviations are given for some of the observation based estimates.*

**Done (Table 3).**

*Line 198: The ACC seasonality is shown in Fig. 8a, not 7a.*

**Done.**

*Line 200: The ACC interannual variability is shown in Fig. 7a, not 8a.*

**Done.**

*Line 249: I don't see a row in Table 3 that would be temperature trend in the Weddell "below 2000 m". Do the authors mean below 3000 m?*

**Revised as suggested.**

*Line 265: I don't see a row in Table 3 that would be temperature trend in the Weddell "between 1000-2000 m". Do the authors mean between 1500 and 2000 m?*

**Revised as suggested.**

*Lines 311-312: Table 3 shows the Ross Sea temperature change at 1500-2000m for B-SOSE (iteration 139) to be less than observations.*

**The relevant sentence is removed in the revised manuscript. We also include additional explanation in Lines 371-373.**

*Line 326: The acronym DSW has not been described up to this point.*

**Done.**

*Line 373: Suggesting adding "previous" before "section".*

**Done.**

*Line 439: Typo, "hydrpgrahic".*

**Done.**

*Line 450: First, thanks to the authors for making all these data and scripts available. I may have missed it, but the link in this line (Nakayama, 2024d) only seemed to include SOSE iteration 105 (didn't see 139).*

**We added Zenodo link for iteration 139.**

*Figure 2c, y-axis label: If there is not enough room to have "March Sea Ice extent", then I suggest changing "March Sea extent" to "March Ice extent".*

**Done.**

*Figure 4 caption: Several of the letter references are incorrect. For example, at the beginning "a,g,l" should be "a,f,k".*

**Done.**

*Figure 5 caption: I don't think the description of the red arrows ("point to the salinity bias") is sufficient. I assume they are pointing to where the largest input of freshwater from Hammond and Jones is introduced and ECCO LLC270 and B-SOSE are the two models that use both the ice shelf melt rate and iceberg calving.*

**We remove these arrows in the revised manuscript.**

*Figure 9 caption: Suggest adding something like "See text for the meaning of the red and blue ellipses in (i)" (or add a description in the figure caption).*

**Done.**

*Figure 11 caption: Suggest changing "blue dots" to "blue dot".*

**Done.**

*Figure 14 caption: Suggest changing "cyan dots" to "cyan dot" (or perhaps "blue dot" since both dots in Figure 1 are the same color).*

**Done.**

*Figure 15 caption: Suggest changing "compiled available all years" to "compiled all available years".*

**Done.**

*Table 3: The figure caption says these are trends "at 500m and 1000 m", but none of the depths given in each row fit that description (all are either 1500-2000m or 3000m-bottom).*

**Thank you for pointing this out. The table caption is fixed.**

---

## Author Comment (AC4)

Author response to the Referee Comments by Reviewer 2 Céline Heuzé on the manuscript:

**Evaluation of MITgcm-based ocean reanalyses for the Southern Ocean**

Yoshihiro Nakayama, Alena Malyarenko, Hong Zhang, Ou Wang, Matthis Auger, Yafei Nie, Ian Fenty, Matthew Mazloff, Köhl Armin, and Dimitris Menemenlis
submitted to Geoscientific Model Developments (https://doi.org/10.5194/egusphere-2024-727)

We thank the Reviewer for the constructive comments. The reviewer's comments are displayed in *italics*, replies are shown in **bold text**.
* * *
*Good paper that will be heavily cited. I have one somewhat-major suggestion to make it even more useful to the overall Southern Ocean community, and a series of minor ones that are mostly esthetic. In the following, the ocean reanalyses and state estimates that are the topic of this paper are referred to as "ORAs".*

*The one major comment: Show us the mixed layer depth. ORAs are too often used instead of observations for both physics and BGC studies when one wants to look at the time evolution of their favourite process (e.g. carbon cycle, primary production). Sure, I am biased in what I read, but this process often relies on a realistic representation of the mixed layers. Besides, it is surprising that you comment on the Weddell Sea hydrography and gyre strength and lament at the representation of both AABW and shelf processes and yet do not check that the mixed layers there are not-too-spurious.*

*I would recommend either a table listing for each ORA, for summer and winter separately, ideally for the Weddell and Ross seas separately, the minimum – median – maximum mixed layer depths, or at least a map of the temporal maximum mixed layer depth for the entire Southern Ocean. The time evolution (for each sea) of the spatial maximum in a style similar to Fig 2 would also be very useful.*

**As suggested, we add two figures (Figures 6 and 7) showing the MLD spatial distributions and timeseries. We also include new text describing our new results (Lines 196-211).**

*Minor comments, in order of appearance: The text contains many typos, starting with the title: ocean reanalysEs, not reanalysIs; and affiliation 5: InsTitute (T missing). I trust a careful reading of the text by the authors and then the copy editors will fix the others.*

**Thank you for pointing this out. We have carefully checked the manuscript before resubmission.**

*Table 1: Add which product used which atmosphere (see later comment), and the year of the bathymetry, since you use this year disparity to explain some of the differences later in the text.*

**As suggested, we include the names of atmospheric reanalysis products in Table 1. We are not able to include the year of bathymetry as this differs depending on regions and observational campaigns. Instead, we include the reference of bathymetric products.**

*In section 2, the methods used to calculate the sea ice extent and volume are not defined. In particular, was a threshold in sea ice concentration used, and if so, which? There is also no information about the sea ice observation source aside from a reference to "satellites" in the figure caption.*

**Thanks for the reminder. We have included the information about the threshold of sea ice concentration and data sources as in section 2.5.**

*Figure 2 onwards: I appreciate that your colour scheme was consistent across figures. I would however recommend that you modify it to increase readability. Red and magenta are hard to distinguish – swapping red for a yellowish orange should work better, and make it compatible with red/green colour-blindness. Dark blue and black are also hard to distinguish – swapping dark blue for a lighter blue such as rgb(0, 150, 255) should work while still remaining distinct enough from green.*

**Done.**

*Figure 3: The observational line is missing. You should have data since the early 2000s at least (Envisat)*

**The observational reference is now added in Figure 3. Two products are used here; one is derived from the Envisat (ES) and CryoSat-2 (CS2) radar altimetry measurements (Hendricks et al., 2018a, 2018b), which covers the period from June 2002 to April 2017. The other was obtained by altimeters on board the satellites SARAL (Ka-band) and CS2 (Ku-band) and covers from May 2013 to October 2018. A detailed description of the observation data has now been added in section 2.5 of the revised manuscript.**

*Line 195: I acknowledge that the main objective of the paper is to describe the performance of the ORAs, not to explain their biases. Yet here and there, you do, relating the biases to bathymetry or shelf processes. This line / entire section presents an opportunity for another one: GECCO3 is systematically weaker than the other ORAs, and it also is the only one that uses NCEP rather than ERA5. Can there be an explanation in the NCEP winds?*

**Done (Line 243-245).**

*Figures 9-10 + 12-13 and Table 3: For panels b,d,f,h,j and the trends, can you please compare them on a similar time period, or at least a similar time span? There could be interannual variability masking or enhancing the trends that you show. Or filter all the data to make sure that you really are showing trends, but that is more effort and not feasible anyway for the short ORAs.*

**The primary purpose of this manuscript is to raise awareness among readers (users) about the importance of model drift when using global ocean simulations. For our analyses, we present two examples from the Weddell and Ross Seas, where model drifts significantly influence the results. These examples illustrate that model drifts can have a substantial impact, making it challenging to distinguish realistic ocean signals from model-induced trends, especially without extensively long model spin-ups. Thus, calculating trends for the same period or filtering all the data may convey a misleading message to readers. We emphasize these points in the revised manuscript (Lines 300-307).**

*Figures 11 and 14: The observational row is missing. You have the data since you show them on Figs 9-10 and 12-13*

**Done.**

*Section 3.5: I have no more comment, but wanted to say that I really liked this section.*

**Thank you!**

---

## Author Response (AR2)

Author response to the Referee Comments by Anonymous Reviewer 1 on the manuscript:

**Evaluation of MITgcm-based ocean reanalyses for the Southern Ocean**

Yoshihiro Nakayama, Alena Malyarenko, Hong Zhang, Ou Wang, Matthis Auger, Yafei Nie, Ian Fenty, Matthew Mazloff, Köhl Armin, and Dimitris Menemenlis

submitted to Geoscientific Model Developments (https://doi.org/10.5194/egusphere-2024-727)

We thank the Reviewer for the constructive comments. The reviewer's comments are displayed in *italics*, replies are shown in **bold text**.
* * *
*General comments*

*I thank the authors for all their efforts in response to my previous comments, especially adding some quantitative comparisons and the mixed layer depth analyses. I have some new comments below, but these are all very minor and I think the authors have produced a nice comparison that will be quite useful (especially the warning about continental shelf properties) to the Southern Ocean community. I'm happy to see this published after the authors have taken the comments below into consideration.*

**Thank you very much for your comments.**

*Specific comments*

*Line 204: I think it would be helpful to compute the mean too shallow MLD bias for each of the models compared to WOA.*

**As suggested, we computed the mean MLDs for all ocean state esimates and WOA (Lines 204-206).**

*Lines 205-211 and Fig. 7: I think this figure is really helpful. My only suggestion is that, since the authors have already computed MLD for different months from WOA for Fig. 6, they add the WOA seasonal cycle of MLD for these two locations in 7c and 7d (similar to what was done for sea ice in 2d and 3d, even though I realize they will not be able to give a variability range for the MLD since WOA is a climatology).*

**In the revised version, I still keep this figure without including timeseries from WOA. All of my plots are either created by plotting WOA directly or plotting ocean observations directly. We do not want to extract data from WOA and plot them because (1) WOA is an interpolated product and (2) there are likely not enough observations to support and evaluate the temporal variability of WOA at these locations.**

*Lines 212-221: Apologies if I missed it, but I never saw it explicitly mentioned what the error in the RMSE is computed with respect to. Is it the WOA values? Also, I know it's discussed later specifically for the Ross and Weddell Gyres, but should there be some mention here of the general Southern Ocean estimated natural variability for the deep temperature and salinity so the reader can see how the variability in the RMSE shown in Figure 8 compares to the observations?*

**The manuscript is revised as suggested (Lines 214-216).**

*Technical corrections*

*Line 42: When the authors state "including GECCO2 (a prior version)", they have not yet introduced GECCO3 and thus I think "a prior version" could be confusing. Suggest expanding the phrase to something like "a prior version of the GECCO3 reanalysis discussed below".*

**Done.**

*Line 59: Suggest changing "improve the Southern Ocean" to either "simulate the Southern Ocean" or "improve the simulation of the Southern Ocean".*

**Done.**

*Lines 108-109 and Table 1: From Verdy and Mazloff (2017), I though B-SOSE iteration 105 used ERA-Interim, not ERA5, for the initial atmospheric forcing.*

**Thank you for pointing this out. The manuscript is revised as suggested.**

*Lines 144-145: Should "small differences between ECCOv4r5" be "small differences between B-SOSE and ECCOv4r5"?*

**Fixed.**

*Line 161: I think "show good agreement at a similar level" should be "show good agreement in February at a similar level".*

**Done.**

*Lines 186-187 and Figure 5: When first looking at Figure 5, it was hard for me to see a fresh bias in the surface salinity between the AP and the Ross for any of the solutions except B-SOSE because of the different color scale for WOA. The contour lines certainly help with this, but I think the different scale for the WOA salinity should be explicitly noted in the Figure 5 figure caption.*

**Thank you for pointing this out. The manuscript is revised as suggested.**

*Lines 197-198: I think the authors should add a reference for the choice of a 0.03 kg/m difference for the definition of MLD.*

**Done.**

*Line 199: Typo, missing "are" between "coincide with or" and "close to".*

**Fixed.**

*Line 331: "Fig. 12" should be "Fig. 13".*

**Fixed.**

*Line 365: I think "presents" should be "present".*

**Fixed.**

*Line 387: Should the location of the Totten Ice Shelf be indicated on some figure?*

**The location of Totten ice shelf is included in Figure 1.**

*Line 453: Should "(Figs. 12,14)" be "(Figs. 12,15)"?*

**Done.**

*Line 492: Suggest putting "However, " at the beginning of the sentence with "MITgcm-based ocean reanalyses...".*

**Done.**

*Line 493: Suggest changing "your research goals" to "their research goals".*

**Done.**

*Figure 5: Did not see the "red and blue ellipses" mentioned in the figure caption.*

**This sentence is removed in the revised manuscript.**

*Captions for Supplementary Figures 3-6: The figures in the main text referred to in these captions have not been updated (e.g. Supp. Fig. 3 caption says "Close-ups of Figs. 9i and 10i" when it should be "12i and 13i".)*

**Thank you for pointing this out. We revise the SI as suggested.**